# Tyramine induces dynamic RNP granule remodeling and translation activation in the Drosophila brain

Nadia Formicola[1], Marjorie Heim[1], Jérémy Dufourt[2], Anne-Sophie Lancelot[1], Akira Nakamura[3], Mounia Lagha[2], Florence Besse[1]*

[1]Université Côte d'Azur, CNRS, Inserm, Institut de Biologie Valrose, Nice, France; [2]Institut de Génétique Moléculaire de Montpellier, University of Montpellier, Montpellier, France; [3]Department of Germline Development, Institute of Molecular Embryology and Genetics, and Graduate School of Pharmaceutical Sciences, Kumamoto University, Kumamoto, Japan

**Abstract** Ribonucleoprotein (RNP) granules are dynamic condensates enriched in regulatory RNA binding proteins (RBPs) and RNAs under tight spatiotemporal control. Extensive recent work has investigated the molecular principles underlying RNP granule assembly, unraveling that they form through the self-association of RNP components into dynamic networks of interactions. How endogenous RNP granules respond to external stimuli to regulate RNA fate is still largely unknown. Here, we demonstrate through high-resolution imaging of intact *Drosophila* brains that Tyramine induces a reversible remodeling of somatic RNP granules characterized by the decondensation of granule-enriched RBPs (e.g. Imp/ZBP1/IGF2BP) and helicases (e.g. Me31B/DDX-6/Rck). Furthermore, our functional analysis reveals that Tyramine signals both through its receptor TyrR and through the calcium-activated kinase CamkII to trigger RNP component decondensation. Finally, we uncover that RNP granule remodeling is accompanied by the rapid and specific translational activation of associated mRNAs. Thus, this work sheds new light on the mechanisms controlling cue-induced rearrangement of physiological RNP condensates.

*For correspondence: besse@unice.fr

Competing interests: The authors declare that no competing interests exist.

## Introduction

Self-assembly of functionally related molecules into the so-called biological condensates has recently emerged as a prevalent process underlying subcellular compartmentalization (*Alberti, 2017*; *Banani et al., 2017*). Condensation of RNA molecules and associated regulatory proteins to form cytoplasmic ribonucleoprotein (RNP) granules, in particular, has been observed in virtually all cell types and species, ranging from bacteria to higher eukaryotes (*Buchan, 2014*; *Cohan and Pappu, 2020*). Different types of RNP granules have been defined based on their composition (e.g. S-foci), function (e.g. P-bodies), origin (e.g. Stress Granules), and/or the cell type they belong to (e.g. germ granules, neuronal granules) (*Kiebler and Bassell, 2006*; *Anderson and Kedersha, 2009*; *Voronina et al., 2011*; *Buchan, 2014*; *De Graeve and Besse, 2018*; *Formicola et al., 2019*). With the notable exception of Stress Granules, most of these RNP granules are found constitutively and have been implicated in the regulation of various aspects of RNA expression, from decay to subcellular RNA localization and translation (*Besse and Ephrussi, 2008*; *Buchan, 2014*; *De Graeve and Besse, 2018*; *Formicola et al., 2019*; *Ivanov et al., 2019*; *Marnik and Updike, 2019*; *Trcek and Lehmann, 2019*). Intriguingly, both repressor and activator functions have been assigned to RNA condensation: clustering of transcripts into translation factories, for example, appears to enhance translation (*Pichon et al., 2016*; *Pizzinga et al., 2019*; *Chouaib et al., 2020*; *Dufourt et al., 2021*), while recruitment of mRNAs to P-bodies, germ granules, or neuronal granules is rather associated

with translational repression (*Krichevsky and Kosik, 2001*; *Fritzsche et al., 2013*; *Hubstenberger et al., 2017*; *Ivanov et al., 2019*; *Kim et al., 2019*; *Tsang et al., 2019*).

Extensive recent work has investigated the molecular principles underlying the assembly and material properties of RNP granules, highlighting the importance of multivalent RNA–RNA, RNA–protein, and protein–protein interactions for demixing of RNP components from the cytoplasm, and formation of phase-separated RNP condensates (*Li et al., 2012*; *Shin and Brangwynne, 2017*; *Mittag and Parker, 2018*; *Van Treeck and Parker, 2018*; *Adekunle and Hubstenberger, 2020*; *Corbet and Parker, 2020*). These studies have also revealed that RNP granule components are not all equivalent. While 'scaffolds' are resident, highly multivalent molecules required for granule assembly, 'clients', in contrast, are dispensable and reversibly recruited by scaffolds (*Banani et al., 2016*; *Ditlev et al., 2018*). Remarkably, the dense yet highly dynamic networks of interactions underlying condensate assembly and maintenance provide the basis for flexible and dynamic compositional changes. Indeed, any alteration in stoichiometry, valency, or binding affinity will dramatically impact on both scaffold condensation and recruitment of client molecules (*Banani et al., 2016*; *Ditlev et al., 2018*; *Sanders et al., 2020*). Because they modulate both protein–protein and protein–RNA interactions, post-translational modifications have in this context been shown to alter the phase behavior of RNA binding proteins (RBPs) in vitro, and to inhibit or promote their partitioning into endogenous condensates in a switch-like manner (*Hofweber and Dormann, 2019*; *Snead and Gladfelter, 2019*).

This framework can in principle explain various aspects of RNP condensate regulation, from assembly to compositional changes or disassembly. Strikingly, the field has until now mostly focused on characterizing the regulatory mechanisms underlying RNP condensate assembly. Understanding how constitutive RNP condensates reversibly reorganize or disassemble in response to distinct stimuli is however equally important. Indeed, dynamic and specific release of condensate-associated mRNAs contributes to spatiotemporal regulation of gene expression in a wide range of physiological and developmental contexts (*Voronina et al., 2011*; *Buchan, 2014*; *Holt et al., 2019*; *Sankaranarayanan and Weil, 2020*). In mature neuronal cells, for example, neuronal RNP granules were shown to control general cell homeostasis, but also to mediate the long-range transport and on-site translation of pre- or post-synaptic mRNAs (*Kiebler and Bassell, 2006*; *De Graeve and Besse, 2018*; *Formicola et al., 2019*). Translational de-repression of granule-associated transcripts has been observed in response to specific neuronal stimuli, and linked to 'RNA unmasking' or dissolution of RNP condensates (*Zeitelhofer et al., 2008*; *Baez et al., 2011*; *Buxbaum et al., 2014*; *Formicola et al., 2019*). To date, however, how neuronal RNP granule components specifically and dynamically re-organize in the context of mature circuits to modulate the expression of associated RNAs is still unclear.

In this study, we uncover that the neuromodulator Tyramine (*Branchek and Blackburn, 2003*; *Burchett and Hicks, 2006*; *Huang et al., 2016*) triggers the reversible remodeling of cytoplasmic RNP granules in the cell bodies of *Drosophila* Mushroom Body (MB) neurons. Through high resolution live imaging of intact brains, we show that this is characterized by the decondensation of two conserved components of RNP granules: the RBP Imp/ZBP1/IGF2BP and the DEAD box helicase Me31B/DDX-6/Rck (*Tiruchinapalli et al., 2003*; *Barbee et al., 2006*; *Miller et al., 2009*; *Hillebrand et al., 2010*; *Vijayakumar et al., 2019*). Functionally, we demonstrate that Tyramine signals through the TyrR receptor and through CamkII, a calcium-activated kinase associating with Imp, to induce RNP granule remodeling. Furthermore, we show that RNP granule remodeling is linked to the translation activation of granule-associated mRNAs, a process we monitor with unprecedented resolution *via* the SunTag amplification system (*Tanenbaum et al., 2014*) recently implemented in *Drosophila* (*Dufourt et al., 2021*). Together, our functional and cellular study sheds new light into the mechanisms underlying signal-induced disassembly of RNP granules. By illustrating how the properties of these macromolecular assemblies can contribute to dynamic and specific regulation of neuronal mRNAs, this work opens new perspectives on the regulation and function of constitutive RNP condensate in physiological contexts.

## Results

### Tyramine induces a reversible remodeling of neuronal RNP granules in MB neurons

In resting brains of 10–15-day-old flies, 100-200 nm-sized cytoplasmic RNP granules are visible in the cell bodies of MB γ neurons (*Figure 1A–A'' and C*), a population of neurons known for its role in learning and memory (*Keene and Waddell, 2007*; *Keleman et al., 2007*; *Akalal et al., 2010*). These granules contain RNAs such as *profilin*, as well as regulatory proteins that dynamically shuttle between the granular and soluble pools (*Vijayakumar et al., 2019*). Among those are the RBP Imp and the DEAD box helicase Me31B, two conserved repressors of translation (*Figure 1A–A''*; *Minshall et al., 2001*; *Nakamura et al., 2001*; *Hüttelmaier et al., 2005*; *Hillebrand et al., 2010*; *Wang et al., 2017*). To investigate the response of neuronal RNP granules to changes in neuronal state, we treated brain explants with different neurotransmitters and neuromodulators known to activate MBs and/or to be involved in learning and memory (*Campusano et al., 2007*; *Martin et al., 2007*; *Majumdar et al., 2012*; *Silva et al., 2015*; *Iliadi et al., 2017*; *Cognigni et al., 2018*; *Sabandal et al., 2020*). The number of Imp-positive RNP granules was scored after 30 min treatment (*Figure 1—figure supplement 1A,B*). Tyramine, a bioamine found in trace amounts in both invertebrate and mammalian brains (*Burchett and Hicks, 2006*; *Lange, 2009*), triggered the strongest response, characterized by the decondensation of Imp molecules and a significant decrease in the number of Imp-containing granules (*Figure 1B,D* and *Figure 1—figure supplement 1C*). Decondensation of Imp was accompanied by a significant, although less pronounced, relocalization of Me31B protein from the granular to the cytoplasmic pool (*Figure 1B'*). This relocalization did not impact on the number of Me31B-positive granules (*Figure 1E*), but translated into a decrease in the ratio between the granular and the cytoplasmic soluble pool of Me31B (partition coefficient; *Figure 1F*). Importantly, the re-localization of Imp and Me31B observed in the presence of Tyramine did not result from changes in protein levels, as similar levels of Imp and Me31B were observed with and without Tyramine treatment (*Figure 1—figure supplement 2*). These results thus indicate that the neuromodulator Tyramine triggers a remodeling of neuronal RNP granules characterized by the differential release of Imp and Me31B RNP components into the cytoplasm.

To test whether granule component decondensation was reversible, we transferred brain explants previously treated for 30 min with Tyramine to regular saline and fixed them after 60 min of recovery. While a significant decrease in the number of Imp-positive granules was observed after Tyramine treatment, bright Imp-positive granules were again visible after the recovery period and their number returned to baseline (*Figure 1—figure supplement 1C*). Similarly, the decreased partitioning of Me31B into granules was reverted after recovery (*Figure 1—figure supplement 1C*). This thus suggests that Tyramine reversibly alters the phase behavior of RNP granule components.

### Dynamics of Tyramine-induced RNP granule remodeling

Although the experiments described above highlighted that neuronal RNP granules dynamically reorganize, they did not provide detailed information about the temporal profile of RNP component decondensation. To monitor in real time the properties of RNP granules, we introduced *via* CRISPR/Cas9 editing a GFP tag in the endogenous *me31B* locus and performed high-resolution real-time imaging of Me31B-GFP-expressing brain explants. This first revealed that granules exhibit a dynamic behavior characterized by successions of short movements and pauses, as well as both fusion and fission events (*Figure 2A* and *Videos 1* and *2*). To dynamically monitor the response of RNP granules to Tyramine, we then imaged Me31B-GFP-positive granules for 30 min after treatment (*Figure 2B–D* and *Videos 3* and *4*) and quantitatively analyzed the partitioning of Me31B over time (*Figure 2B*). This revealed that relocalization of Me31B from the granular to the cytoplasm pool is initiated within minutes after Tyramine treatment, but is a progressive rather than abrupt process. As shown in *Figure 2—figure supplement 1*, a similar trend was observed for GFP-Imp, illustrated by a progressive decrease in the number of GFP-Imp-positive granules.

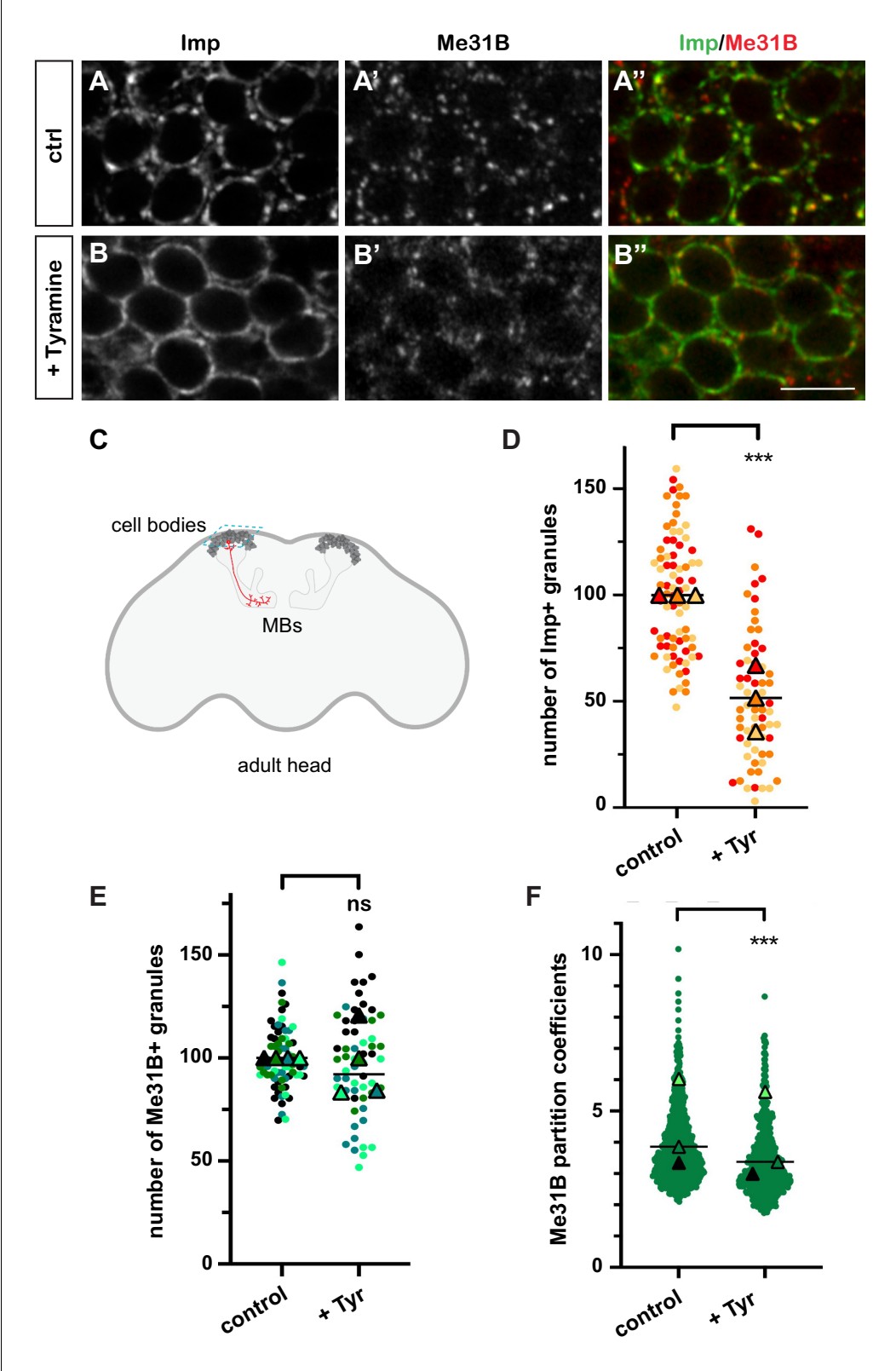

**Figure 1.** Neuronal ribonucleoprotein (RNP) granules undergo remodeling in response to Tyramine. (A, B) Cell bodies of adult Mushroom Body (MB) γ neurons stained with anti-Imp (A, B, green in A', B') and anti-Me31B (A', B', red in A', B') antibodies. Brain explants were incubated for 30 min in saline (A–A', control) or in saline supplemented with 10 mM Tyramine (B–B"). Note that most of the cell body volume is occupied by the nucleus, and thus that the cytoplasm is visible as a ring on confocal sections. Scale bar in A, B: 5 µm. (C) Schematic representation of an adult *Drosophila* head, with MBs

*Figure 1 continued on next page*

*Figure 1 continued*

highlighted. The morphology of a single MB γ neuron is represented in red. The region imaged to analyze RNP granule behavior is boxed (turquoise dotted lines). (D, E) Normalized numbers of Imp- (D) or Me31B- (E) containing granules (per image field). Individual data points were color-coded based on the experimental replicate they belong to. Three (D) to four (E) replicates were performed and the mean value of each replicate is indicated as a symbol (triangle). At least 20 (D) or 12 (E) data points were collected for each replicate. ***, p<0.001 (t-test on individual data points). n.s. stands for not significant. (F) Distribution of Me31B partition coefficients. Partition coefficients were estimated by dividing the intensity of Me31B signal in individual RNP granules to the intensity of the cytoplasmic diffuse pool (see Materials and methods) and calculated for each granule detected in the imaged fields. The individual data points displayed on the graph were extracted from a single replicate. Three replicates were performed and the mean value of each is indicated as a symbol (triangle). Number of RNP granules: 622 granules distributed across 18 fields (control), 777 granules distributed across 18 fields (+ Tyramine). ***, p<0.001 (t-test on individual data points). Note that the p-value obtained when comparing the distributions of replicate means is 0.7 (Mann–Whitney test). For the list of values used to generate the graphs shown in D–F see *Figure 1—source data 1*.

The online version of this article includes the following source data and figure supplement(s) for figure 1:

**Source data 1.** Numerical data to support graphs in *Figure 1D-F*.
**Figure supplement 1.** Quantification and detection of ribonucleoprotein (RNP) granule number in response to neurotransmitters and reversibility.
**Figure supplement 1—source data 1.** Numerical data to support graphs in *Figure 1—figure supplement 1*.
**Figure supplement 2.** Quantification of Me31B and Imp protein levels after Tyramine treatment.
**Figure supplement 2—source data 1.** Numerical data to support graph in *Figure 1—figure supplement 2B*.
**Figure supplement 2—source data 2.** Original images to support *Figure 1—figure supplement 2A*.

## Tyramine signals through the TyrR receptor to induce neuronal activation and RNP component decondensation

Having discovered the impact of Tyramine on RNP granules, we next wondered how Tyramine signaling was mediated. A number of receptors responding to Tyramine have been identified in *Drosophila* (*Ohta and Ozoe, 2014*; *El-Kholy et al., 2015*), yet only one – the GPCR TyrR – has been shown to respond specifically to Tyramine, and not to other biogenic amines (*Cazzamali et al., 2005*; *Huang et al., 2016*). To thus test whether TyrR would mediate neuronal RNP granule remodeling, we treated *TyrR^Gal4* null mutant brain explants with Tyramine and analyzed the behavior of granule markers. As shown in *Figure 3A–E*, decondensation of both Imp and Me31B was significantly impaired upon *TyrR* inactivation, suggesting that TyrR is the main receptor involved.

Both neuromodulatory and neurotransmitter functions have been assigned to Tyramine to date (*Nagaya et al., 2002*; *Pirri et al., 2009*; *Huang et al., 2016*; *Jin et al., 2016*; *Schützler et al., 2019*). To investigate whether Tyramine induced a calcium response in MB neurons, we monitored calcium transients upon exposure of brain explants to Tyramine, using a genetically encoded calcium indicator expressed in MBs (MB247-homer::GCamp3.0; *Pech et al., 2015*). Tyramine elicited calcium transients peaking 2–8 min after exposure (*Figure 3F* and *Videos 5* and *6*), as well as a modest, but reproducible and dose-dependent, long-term increase in intracellular $Ca^{2+}$ (*Figure 3G*). Importantly, inactivation of *TyrR* largely (although not completely) inhibited the main calcium peak induced by Tyramine (*Figure 3F*), confirming the specificity of MB neuronal response. The slow response observed upon Tyramine exposure suggests that Tyramine may activate MB neurons indirectly through other neurons. Consistent with this idea, inhibiting the firing of MB neurons through conditional expression of the inward-rectifying potassium channel Kir 2.1 (*Paradis et al., 2001*) significantly suppressed Tyramine-induced Imp decondensation (*Figure 1—figure supplement 1D*), indicating the need for both evoked responses and GPCR-mediated signaling and suggesting the existence of TyrR-expressing neurons acting as a relay to transduce Tyramine signaling.

## CamkII is required for Tyramine-induced RNP granule remodeling

To identify the proteins involved in the dynamic regulation of RNP granules, we expressed GFP-tagged Imp proteins specifically in MB γ neurons and immunoprecipitated GFP-fusions from adult head lysates (*Figure 4—figure supplement 1A* and Materials and methods). Co-precipitated proteins were identified by mass spectrometry, and heads expressing sole GFP were used as a specificity control. In total, 51 proteins were reproducibly identified, distributed into various functional categories (*Figure 4—figure supplement 1B* and *Supplementary file 1*). As expected, RBPs were strongly enriched in the bound fraction (22/51 proteins; p<0.001). Not all RBPs present in Imp RNP granules were however recovered (*Vijayakumar et al., 2019*), presumably because our immunoprecipitation approach mainly targeted soluble cytoplasmic complexes. Among the identified Imp

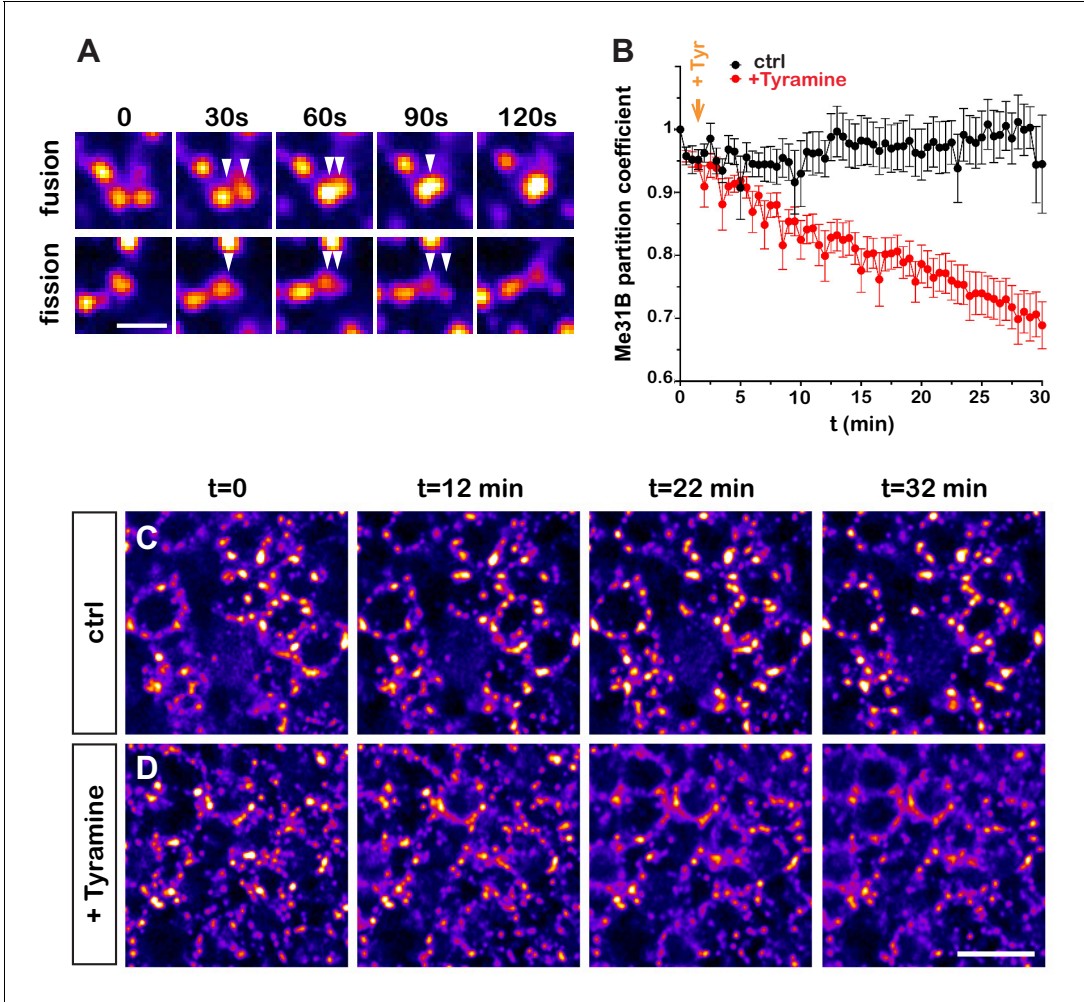

**Figure 2.** Dynamics of Tyramine-induced ribonucleoprotein (RNP) granule remodeling. (**A**) Image sequences extracted from movies following Me31B-GFP-positive granules over time. Both fusion (upper panel) and fission (lower panel) events are shown. GFP intensities are represented using the 'Fire' LUT of ImageJ. Scale bar: 2 µm. (**B**) Me31B-GFP mean partition coefficients in function of time in brain explants treated (red) or not (black) with 10 mM Tyramine. Each data point represents the mean of the average partition coefficients measured for all granules present in fields imaged at a given time point. Tyramine was added at t = 2 min (orange arrow). (**C, D**) Image sequences extracted from movies recording the cell bodies of adult Mushroom Body (MB) γ neurons endogenously expressing Me31B-GFP proteins. Brain explants were either maintained in saline (**C**), or supplemented with 10 mM Tyramine (**D**) at t = 2 min. Images were originally acquired every 30 s. Intensities are displayed using the 'Fire' LUT of ImageJ. Scale bar: 5 µm. Numbers of movies: 7 (ctrl) and 11 (+ Tyr). Note that in these experiments MB γ neurons could not be unambiguously distinguished from other MB neuronal subpopulations. No difference could however be observed in the behavior of Me31B-GFP-positive granules within MB neurons. For the list of values used to generate the graphs shown in B see *Figure 2—source data 1*.

The online version of this article includes the following source data and figure supplement(s) for figure 2:

**Source data 1.** Numerical data to support graphs in *Figure 2B*.

**Figure supplement 1.** Mean numbers of GFP-Imp-positive granules in function of time in brain explants.

**Figure supplement 1—source data 1.** Numerical data to support graphs in *Figure 2—figure supplement 1*.

interactors, we focused our attention on Ca$^{2+}$/calmodulin-dependent protein kinase II (CamkII), as it is a conserved kinase activated in response to calcium rises (*Coultrap and Bayer, 2012*). To validate the association between Imp and CamkII, we performed co-immunoprecipitation experiments in cultured S2R+ cells. As shown in *Figure 4A*, CamkII co-immunoprecipitated with GFP-Imp, but not with sole GFP. Furthermore, CamkII interacted with Imp both in the presence and in the absence of RNase, indicating that the Imp/CamkII interaction is RNA-independent. In vivo, both CamkII and phospho-CamkII (the active form of CamkII) were found diffusely localized in the cytoplasm of MB γ

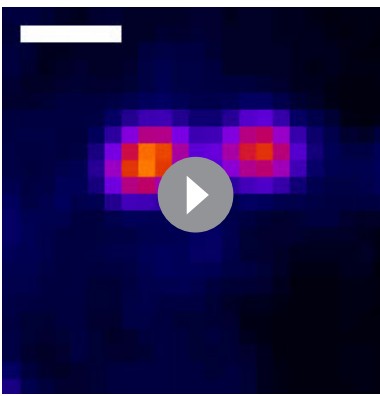

**Video 1.** Fusion between two Me31B-GFP-containing granules. Real-time imaging of Me31B-GFP-containing granules in the cell body of an intact adult brain explant. Signal intensities are displayed using the 'Fire' LUT of ImageJ. Images were acquired every 30 s. Scale bar: 0.5 μm.

https://elifesciences.org/articles/65742#video1

neurons, similar to the soluble pool of Imp molecules (*Figure 4—figure supplement 1C*). No particular enrichment of CamkII was observed in Imp-containing RNP granules.

To then investigate whether the function of CamkII was important for the remodeling of neuronal RNP granules in response to Tyramine, we expressed in MB neurons the ala peptide, a peptide derived from the CamkII autoinhibitory segment that binds to the catalytic site and was shown to selectively inhibit CamkII activity, both in vitro and in vivo (*Griffith et al., 1993*; *Carrillo et al., 2010*; *Nesler et al., 2016*; *Newman et al., 2017*). Conditional expression of the ala peptide significantly inhibited the decondensation of Imp molecules observed upon Tyramine treatment (*Figure 4B*), demonstrating that CamkII is required cell-autonomously downstream of Tyramine to promote Imp decondensation. Ala-mediated inhibition of CamkII, however, did not inhibit the partial cytoplasmic relocalization of Me31B observed in response to Tyramine (*Figure 4C*), indicating that CamkII specifically modulates the decondensation of Imp.

## Tyramine induces the translational activation of granule-associated mRNAs

Neuronal RNP granules are thought to maintain associated mRNAs in a translationally silenced state (*Krichevsky and Kosik, 2001*; *Fritzsche et al., 2013*; *El Fatimy et al., 2016*; *De Graeve and Besse, 2018*). To investigate whether the observed release of granule components is accompanied by the translational derepression of granule-associated mRNAs, we monitored the translation of *profilin*, an mRNA known to be directly bound by Imp and present in MB RNP granules (*Medioni et al., 2014*; *Vijayakumar et al., 2019*). First, we analyzed the expression of a reporter in which the 3'UTR of *profilin* is fused to the coding sequence of EGFP. Constructs generated with the *SV40* 3'UTR were used as a negative control. As shown in *Figure 5A*, treating brains with Tyramine induced a significant increase in GFP signal intensity for the construct containing *profilin* 3'UTR, but not for that containing *SV40* 3'UTR. Furthermore, no significant increase in GFP expression was observed upon prior incubation of *gfp-profilin 3'UTR*-expressing brains with the translation inhibitor anisomycin (*Figure 5B*), indicating that increased GFP levels result from increased protein synthesis. To extend our analysis to other mRNAs, we then monitored the response of two other reporters: one enriched in Imp and Me31B-positive granules in control conditions (*gfp-cofilin* 3'UTR), the other not (*gfp-camk2* 3'UTR) (K. Pushpalatha and F. Besse, unpublished). Remarkably, increased expression of *gfp-cofilin* 3'UTR, but not of *gfp-camk2* 3'UTR reporters, was observed upon Tyramine treatment (*Figure 5A*), suggesting that translation activation is specific to granule-associated mRNAs. Further consistent with a model where RNP granule remodeling relieves associated mRNAs from translational repression, blocking Imp decondensation through inhibition of

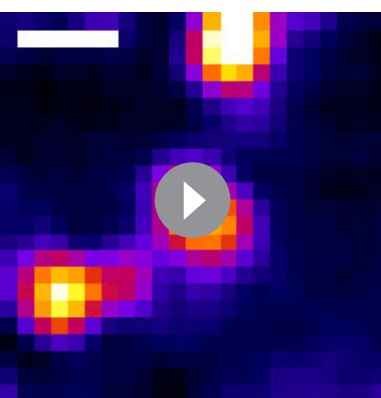

**Video 2.** Fission of a Me31B-GFP-containing granule. Real-time imaging of Me31B-GFP-containing granules in the cell body of an intact adult brain explant. Signal intensities are displayed using the 'Fire' LUT of ImageJ. Images were acquired every 30 s. Scale bar: 0.5 μm.

https://elifesciences.org/articles/65742#video2

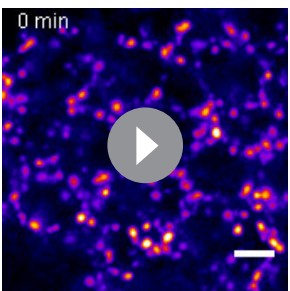

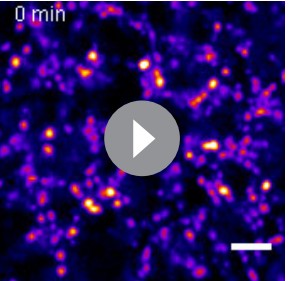

**Video 3.** Behavior of Me31B-GFP-containing granules in brain explants treated with control saline. Real-time imaging of MB γ cell bodies expressing endogenously expressed Me31B-GFP. Signal intensities are displayed using the 'Fire' LUT of ImageJ. Images were acquired every 30 s for 30 min. Control HL3 buffer was added at t = 2 min. Scale bar: 3 μm.

https://elifesciences.org/articles/65742#video3

**Video 4.** Dynamic response of Me31B-GFP- containing granules to Tyramine. Real-time imaging of MB γ cell bodies expressing endogenously expressed Me31B-GFP. Signal intensities are displayed using the 'Fire' LUT of ImageJ. Images were acquired every 30 s for 30 min, from intact adult brain explants. Tyramine was added at t = 2 min to reach a 10 mM final concentration. Scale bar: 3 μm.

https://elifesciences.org/articles/65742#video4

CamkII prevented the translational de-repression of *profilin* 3'UTR reporters (*Figure 5C*).

As GFP-based reporters reflect translation status indirectly, and with poor temporal dynamics, we then aimed at monitoring translation activation with high spatiotemporal resolution, using the Sun-Tag methodology (*Pichon et al., 2016*; *Wang et al., 2016*; *Wu et al., 2016*; *Yan et al., 2016*), recently deployed in *Drosophila* (*Dufourt et al., 2021*). SunTag-tagged *profilin* transcripts were co-expressed in MB γ neurons together with scFv-GFP-NLS fusions to detect translation sites and brains were imaged in real-time. Strikingly, Tyramine induced within minutes the formation of bright GFP-positive cytoplasmic foci (*Figure 6A* and *Video 7*). These foci were not observed in the absence of SunTag-tagged transcripts (*Figure 6A,B* and *Video 8*). Furthermore, their formation was inhibited by puromycin (*Figure 6A,B* and *Video 9*), indicating that they form through translation and likely correspond to translation foci. Remarkably, activation of *profilin* translation occurred as a burst, with kinetics very similar to that of Tyramine-induced calcium transients. As shown in *Figure 6C* and *Video 7*, indeed, the number of cells exhibiting SunTag foci peaked 2–8 min after Tyramine exposure, before progressively reverting to baseline values.

As translation peaked before RNP components completed their decondensation, we wondered if mRNAs would be released from granules rapidly after Tyramine treatment. smFISH experiments were thus performed 10 min after treatment to monitor the association of endogenous *profilin* transcripts (*Figure 5—figure supplement 1A,B*), or *gfp-profilin* 3'UTR transcripts (*Figure 5D* and *Figure 5—figure supplement 1C–E*) with RNP granules. Both experiments revealed a significant decrease in the number of *profilin* mRNA or reporter RNA contained in RNP granules, indicating that release of mRNAs represents an early step of granule remodeling that temporally matches with translation activation.

## Discussion

### Tyramine triggers RNP component decondensation and translation activation

Membrane-less RNP condensates enriched in transcripts under tight regulatory control, as well as regulators of RNA translation, transport or decay have been described in various cell types and organisms (*Buchan, 2014*). Neurons exhibit a particularly complex collection of RNP granules composed of both shared and distinct components, raising the question of how granule composition is established and dynamically regulated (*Fritzsche et al., 2013*; *De Graeve and Besse, 2018*; *Formicola et al., 2019*). Frameworks have been proposed to explain RNP granule compositional control, in which scaffold molecules (or nodes) establish a core network of multivalent interactions essential for both granule nucleation and further recruitment of more dynamically associated client

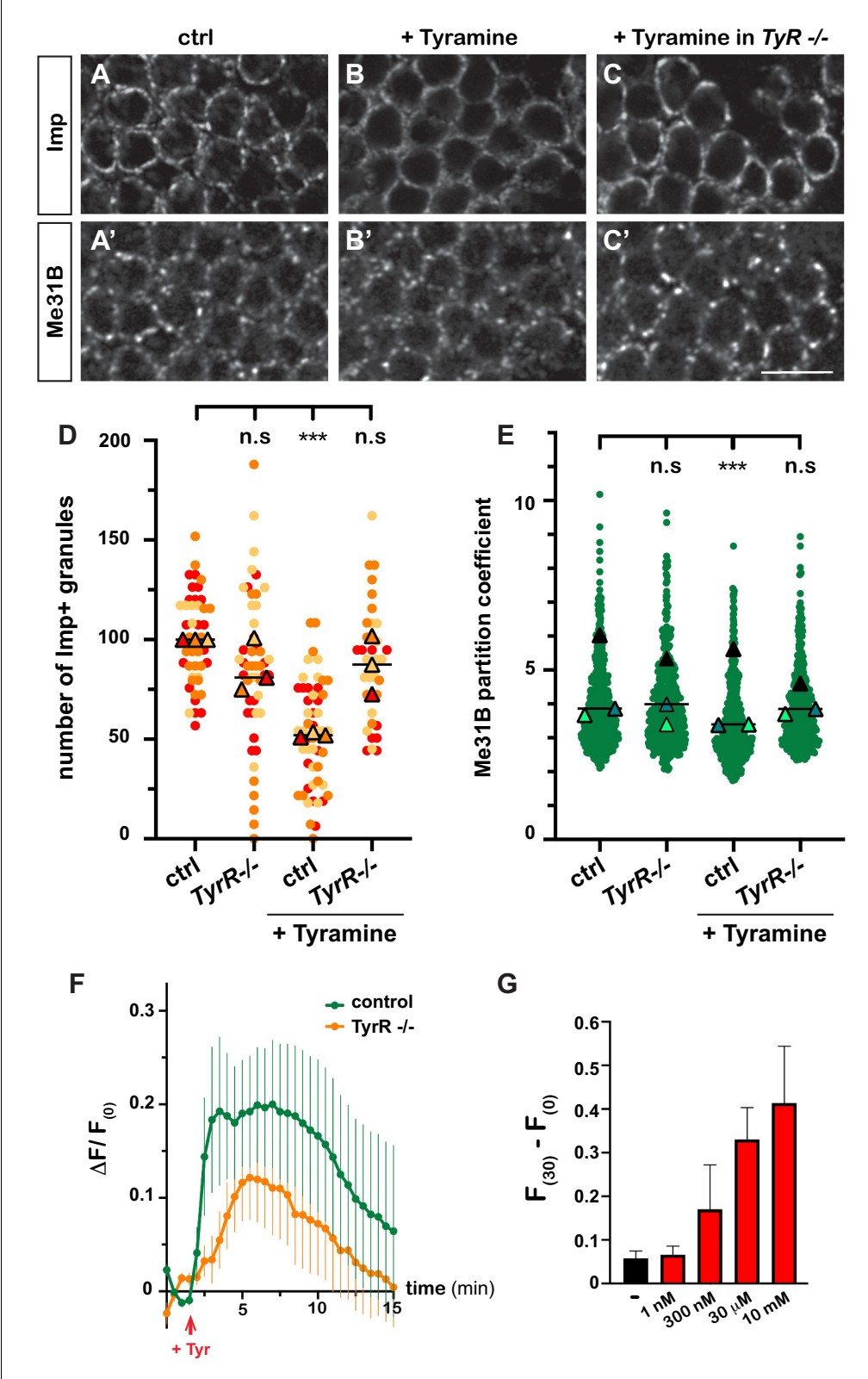

**Figure 3.** Tyramine induces TyrR-dependent responses in MB neurons. (**A–C**) Cell bodies of control (**A, B**) or *TyrR^Gal4* mutant (**C**) adult Mushroom Body γ neurons stained with anti-Imp (**A–C**) and anti-Me31B (**A'–C'**) antibodies. Brains were incubated in saline (**A, A'**, control) or in saline supplemented with 10 mM Tyramine (**B–C'**) Scale bar: 5 μm. (**D**) Normalized numbers of Imp-containing granules (per image field). Individual data points were color-coded based on the experimental replicate they belong to. Three replicates were performed for each condition and the mean value of each replicate is

*Figure 3 continued on next page*

*Figure 3 continued*

indicated as a symbol (triangle). At least 10 data points were collected for each replicate. ***, p<0.001 (Kruskal–Wallis test on individual data points with Dunn's post-test). n.s. stands for not significant. (**E**) Distribution of Me31B partition coefficients. Partition coefficients were estimated by dividing the intensity of Me31B signal in individual ribonucleoprotein (RNP) granules to the intensity of the cytoplasmic diffuse pool and calculated for each granule detected in the imaged fields. The individual data points displayed on the graph were extracted from a single replicate. Three replicates were performed and the mean value of each is indicated as a symbol (triangle). At least 10 fields were analyzed per condition. Number of RNP granules: 875 granules distributed across 18 fields (control), 558 granules distributed across 14 fields (+ Tyramine), and 475 granules distributed across 10 fields (+ Tyramine in *TyrR$^{Gal4}$* mutants). ***, p<0.001 (one-way ANOVA on individual data points with Dunnett's post-tests). n.s. stands for not significant. (**F**) Average fluorescence intensity (F) of GCamp3.0 signal in MB calyx upon exposure of control (green) or *TyrR$^{Gal4}$* mutants (orange) brain explants to 10 mM Tyramine. The MB247-homer::GCamp3.0 reporter was used to monitor Ca$^{2+}$ levels. Data are plotted as F(t)-F(t = 0)/F(t = 0) (ΔF/F(0); see Materials and methods). Tyramine was added at t = 2 min (red arrow). Error bars correspond to S.E.M. Number of brains analyzed: 18 (control) and 13 (*TyrR$^{Gal4}$* mutants). (**G**) Dose-dependent long-term increase in Ca$^{2+}$ levels upon exposure to Tyramine. Intensities of GCamp3.0 signal are plotted as F(t = 30 min) – F(t = 0). Numbers of brains analyzed: 5 (-), 5 (1 nM), 6 (300 nM), 6 (30 μM), and 8 (10 mM). For the list of values used to generate the graphs shown in D–G see *Figure 3—source data 1*.

The online version of this article includes the following source data for figure 3:

**Source data 1.** Numerical data to support graphs in *Figure 3D-G*.

molecules (*Banani et al., 2016*; *Ditlev et al., 2018*; *Sanders et al., 2020*). In this model, modulation of scaffold or client partitioning properties can lead to the differential release of granule-enriched proteins and RNAs. Whether and how these frameworks apply to endogenous neuronal RNP granules has remained unclear. Our results uncovered that Tyramine stimulation differentially impacts on neuronal RNP components, as a nearly complete release of granule-associated Imp, but only a partial relocalization of granular Me31B was observed in response to Tyramine. These results, together with our observation that inactivation of *me31B*, but not of *imp*, prevents granule assembly (K. Push-palatha and F. Besse, unpublished) are consistent with a model in which Me31B behaves as a scaffold, and Imp as a client whose partitioning into granules depends on Me31B, and is specifically modulated by Tyramine. Interestingly, the release of Imp from neuronal granules is associated with

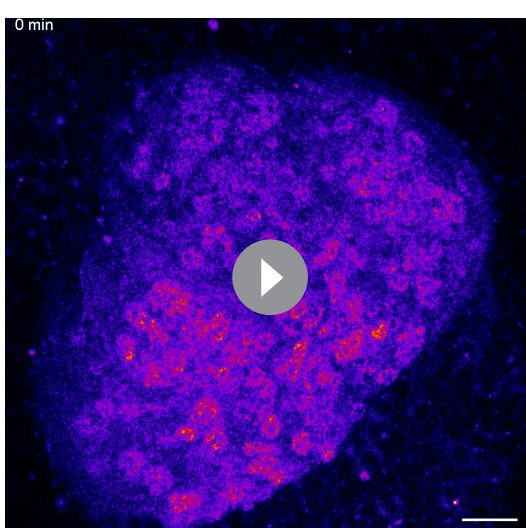

**Video 5.** Tyramine induces a rise in the intracellular Ca$^{2+}$ concentration of control MB neurons. Real-time imaging of MB neurons (calyx region) expressing a Homer::GCamp3.0 construct under the control of the MB247 promoter. Signal intensities are displayed using the 'Fire' LUT of ImageJ. Images were acquired every 30 s for 15 min, from intact adult brain explants. Tyramine was added at t = 2 min. Scale bar: 10 μm.
https://elifesciences.org/articles/65742#video5

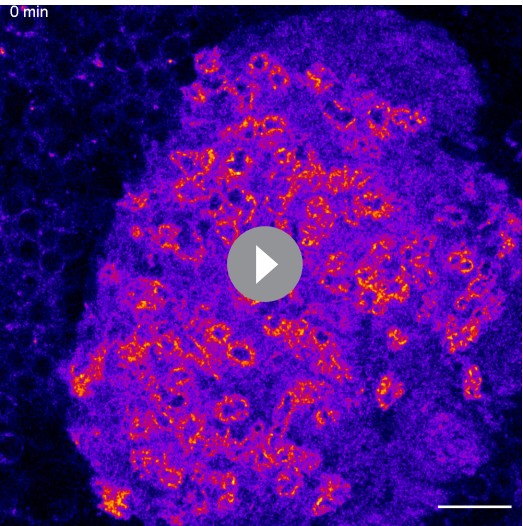

**Video 6.** Tyramine-induced calcium response is inhibited in TyrR-/- mutant context. Real-time imaging of a *TyrR$^{Gal4}$* brain in which MB neurons expressed a Homer::GCamp3.0 construct under the control of the MB247 promoter. MB calyx region is shown. Signal intensities are displayed using the 'Fire' LUT of ImageJ. Images were acquired every 30 s for 15 min, from intact adult brain explants. Tyramine was added at t = 2 min. Scale bar: 10 μm.
https://elifesciences.org/articles/65742#video6

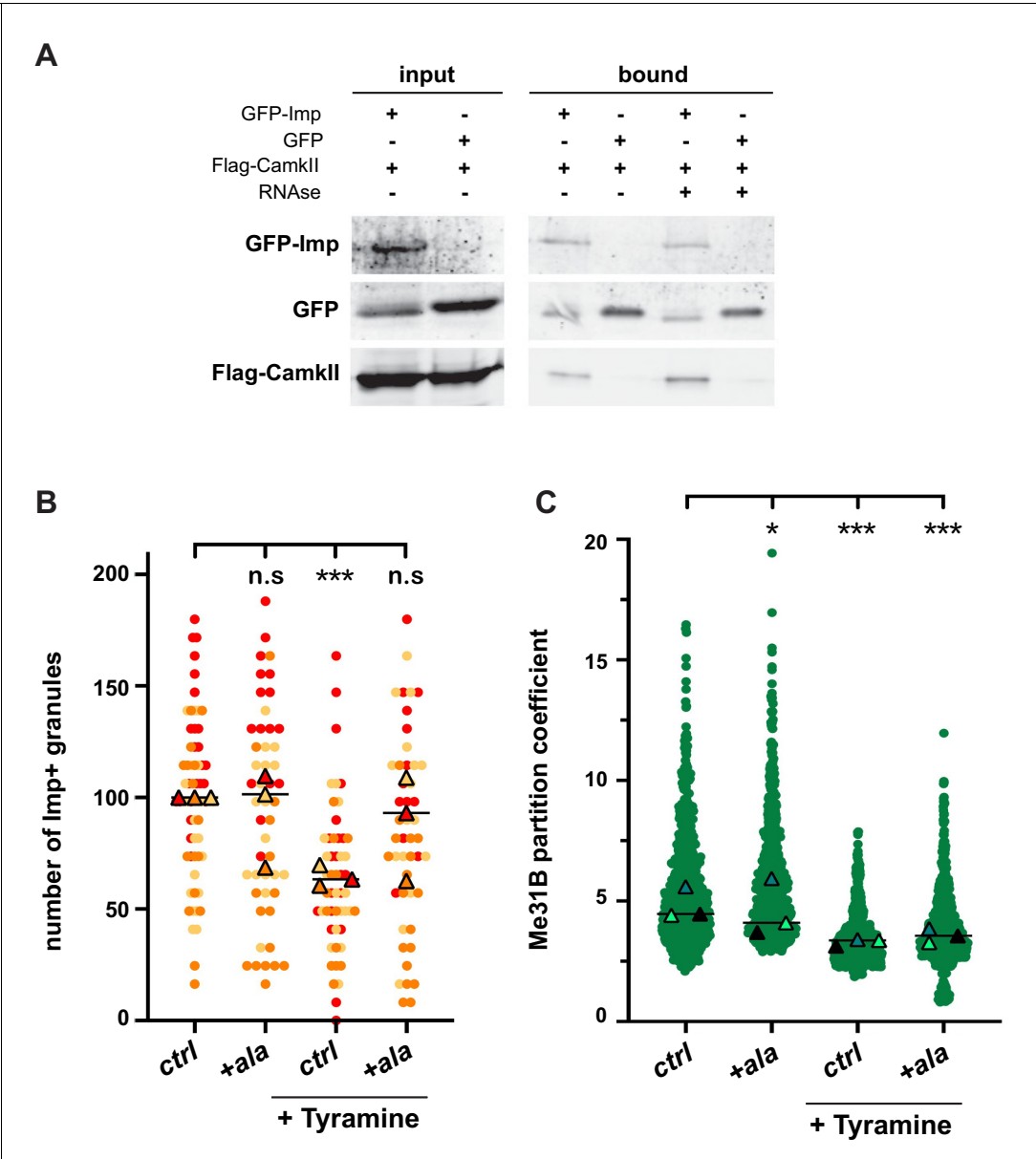

**Figure 4.** CamkII interacts with Imp and is required for Tyramine-induced Imp decondensation. (**A**) CamkII co-immunoprecipitates with Imp. FLAG-CamkII constructs were co-transfected with either GFP-Imp or GFP (negative control) in S2R+ cells. GFP proteins were immunoprecipitated and the bound fractions (right) used for western blot. Input fractions (left) were used as a control of expression. Anti-FLAG and anti-GFP antibodies were used to detect respectively CamkII (MW ≈ 55–60 kDa) and Imp (MW ≈ 90 kDa) fusion proteins. Cell lysates were treated (+) or not (−) with RNase prior to immunoprecipitation. (**B**) Normalized numbers of Imp-containing granules in brain explants treated (+ Tyramine) or not (ctrl) with 10 mM Tyramine. + ala refers to the condition where the CamkII inhibitory peptide ala was expressed specifically in MB neurons, using the tub-Gal80ts;;OK107-Gal4 driver. Individual data points were color-coded based on the experimental replicate they belong to. Three replicates were performed for each condition and the mean value of each replicate is indicated as a symbol (triangle). At least 12 data points were collected for each replicate. (**C**) Distribution of Me31B partition coefficients. Me31B partition coefficients were estimated by dividing the intensity of Me31B signal in individual ribonucleoprotein (RNP) granules to the intensity of the cytoplasmic diffuse pool and calculated for all the granules detected in imaged fields. The individual data points displayed on the graph were extracted from a single replicate. Three replicates were performed and the mean value of each is indicated as a symbol (triangle). Number of RNP granules: 622 granules distributed across 18 fields (control), 777 granules distributed across 18 fields (+ Tyramine) and 663 granules distributed across 16 fields (+ Tyramine +ala). *, p<0.05; ***, p<0.001 (one-way ANOVA on individual data points with Dunnett's post-tests). n.s. stands for not significant. For the list of values used to generate the graphs shown in B, C see *Figure 4—source data 1*. For original western blot images see *Figure 4—figure supplement 1—source data 1*.

The online version of this article includes the following source data and figure supplement(s) for figure 4:

**Source data 1.** Numerical data to support graphs in *Figure 4B, C*.

*Figure 4 continued on next page*

*Figure 4 continued*

**Source data 2.** Original images to support *Figure 4A*.
**Figure supplement 1.** CamkII interacts with Imp.
**Figure supplement 1—source data 1.** Numerical data to support graphs in *Figure 4—figure supplement 1D*.

the translational activation of its target RNA *profilin*, suggesting that differential release of client RBPs, rather than complete granule disassembly, might represent a means to modulate the expression of specific sets of client-associated RNAs in response to distinct stimuli. As further revealed by our real-time imaging experiments, translation activation and granule component decondensation, although both starting within minutes after Tyramine exposure, exhibit distinct temporal profiles. While translation activation occurs as a burst, decondensation of RNP components is a continuous and progressive process. This may result from the gradual depletion of the pool of translationally repressed mRNAs normally dynamically recruited to RNP granules.

## CamkII activity is required downstream of Tyramine to inhibit Imp partitioning

Phosphorylation is a rapid and reversible post-translational modification that dramatically impacts on both intra- and inter-molecular interactions. Not surprisingly then, phosphorylation was shown in vitro and in cells to regulate the partitioning of RNP components into condensates, either positively or negatively depending on the context (*Hofweber and Dormann, 2019*; *Kim et al., 2019*). In living systems, studies have so far pointed to a role for granule-enriched kinases in the phosphorylation of scaffold proteins, leading to granule disassembly. This process is for example required for clearance of Stress Granule upon recovery (*Wippich et al., 2013*; *Krisenko et al., 2015*; *Shattuck et al., 2019*), or for dissolution of anteriorly localized P-granules in *C. elegans* zygotes (*Wang et al., 2014*). If and how phosphorylation modulates the recruitment of client molecules to regulate RNP composition rather than assembly/disassembly has so far remained poorly explored. Here, we showed that the calcium-activated CamkII kinase interacts with Imp, and is required downstream of Tyramine to trigger Imp decondensation. As Tyramine induces a calcium response in MB neurons, our results thus suggest a model in which Tyramine-induced activation of CamkII may promote its association with Imp, thus inhibiting the partitioning of Imp into somatic RNP granules (*Figure 6—figure supplement 1*). Whether CamkII directly phosphorylates Imp remains to be investigated. Although Imp contains four CamkII consensus phosphorylation sites (RXX<u>S/T</u>), mutating these four sites into RXXA by CRISPR/Cas9-mediated engineering did not prevent the decondensation of phosphomutant Imp proteins in response to Tyramine (*Figure 4—figure supplement 1D*). CamkII may thus either phosphorylate Imp through non-canonical sites, as described for other targets (*Kennelly and Krebs, 1991*; *Sun et al., 1994*; *Huang et al., 2011*; *Herren et al., 2015*), or phosphorylate partner(s) of Imp essential for its association with neuronal RNP granules. Interestingly, our results suggested that CamkII is not enriched within RNP granules, raising the hypothesis that kinases may modulate RNP granule composition by targeting the soluble pool of client molecules, rather than the granule-associated pool.

## Tyramine signals through the TyrR receptor to activate MB neurons and trigger RNP granule remodeling

Tyramine is a biogenic amine produced from Tyrosine in neurons expressing the Tyrosine decarboxylase enzyme (*Lange, 2009*). Although it has for long exclusively been considered as a precursor of Octopamine, recent work has revealed Tyramine-dependent but Octopamine-independent function in *Drosophila*, identifying Tyramine as a neuroactive chemical modulating different aspects of animal physiology and behavior (*Huang et al., 2016*; *Schützler et al., 2019*). A number of GPCRs responsive to Tyramine have been cloned and pharmacologically tested, and three have been defined as Tyramine receptors (Oct-Tyr, TyrR, and TyrRII). Only one was shown to bind Tyramine with high affinity and specificity: TyrR (also termed CG7431 or TAR2) (*Cazzamali et al., 2005*; *Ohta and Ozoe, 2014*). As indicated by single-cell transcriptomic analyses, neither TyrR nor any of the other Tyramine receptors is significantly expressed in MB neurons (*Davie et al., 2018*). Our results, however, have shown that both the calcium response and the remodeling of neuronal RNP granules triggered in

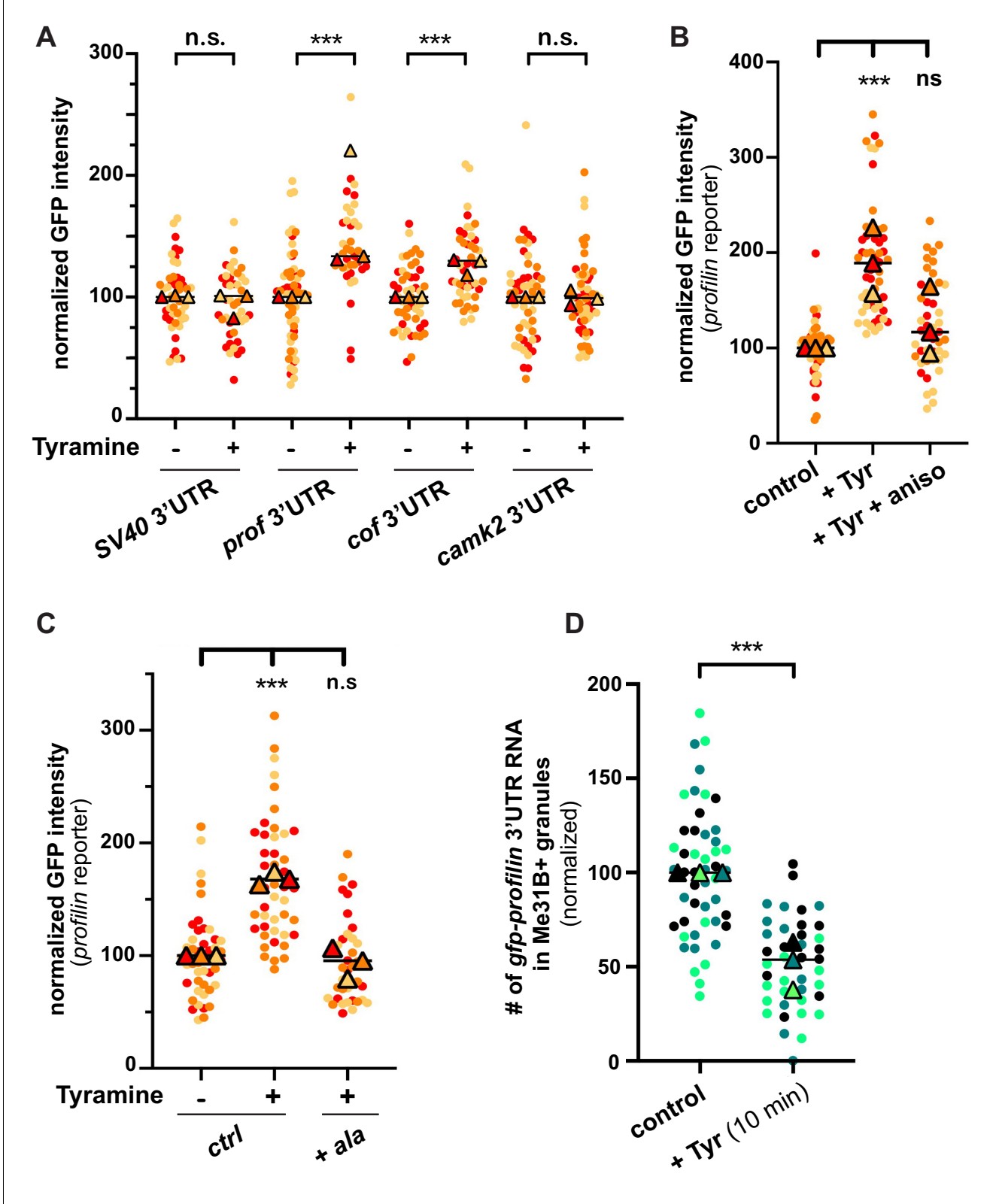

**Figure 5.** The translation of granule-associated mRNAs is increased upon Tyramine treatment. (**A**) Normalized GFP signal intensities produced by reporters in which the 3'UTR of different transcripts was fused to GFP. (**B**) Normalized GFP signal intensities produced by the GFP-*profilin* 3'UTR reporter. In **A** and **B**, reporters were expressed for 3 days under the control of tub-Gal80ts;;OK107-Gal4. Brain explants were treated (+Tyr) or not (ctrl) with 10 mM Tyramine for 30 min. In **B**, anisomycin (aniso) was added in addition to Tyramine to block translation. For the *profilin* 3'UTR reporter in **A**,

*Figure 5 continued on next page*

*Figure 5 continued*

three outlier data were omitted from the graph (although they were considered to calculate the mean of the corresponding replicate and to perform statistical tests). (C) Normalized GFP signal intensities produced by the GFP-*profilin* 3'UTR reporter. GFP-*profilin* 3'UTR was expressed solely (ctrl), or together with ala (+ ala), and brain explants were treated (+) or not (−) with 10 mM Tyramine for 30 min. The ala inhibitory peptide was expressed conditionally in adult MB neurons using tub-Gal80ts;;OK107-Gal4. (D) Proportion of *gfp-profilin* 3'UTR RNA molecules contained in Me31B-mTomato-positive granules. Co-localization was measured using the JACoP plugin of ImageJ (see Materials and methods and *Figure 4—figure supplement 1C*) and values normalized to controls. Individual data points in A–D were color-coded based on the experimental replicate they belong to. Three replicates were performed for each condition and the mean value of each replicate is indicated as a symbol (triangle). At least 10 data points were collected for each replicate. ***, p<0.001 (t-tests on individual data points for A, D and Kruskal–Wallis test on individual data points with Dunn's post-tests for B, C). n.s. stands for not significant. For the list of values used to generate the graphs shown in A–D see *Figure 5—source data 1*.

The online version of this article includes the following source data and figure supplement(s) for figure 5:

**Source data 1.** Numerical data to support graphs in *Figure 5*.

**Figure supplement 1.** Release of endogenous *profilin* mRNA from granules after 10 min Tyramine treatment.

**Figure supplement 1—source data 1.** Numerical data to support graphs in *Figure 5—figure supplement 1*.

MB neurons by Tyramine depend on the function of TyrR. This thus suggests that Tyramine may act on MB neurons indirectly, through TyrR-expressing neurons innervating MBs (*Figure 6—figure supplement 1*). Consistent with this idea, the calcium response we observed in MBs in response to Tyramine is slow, peaking 2–8 min after addition of Tyramine. Furthermore, Tyramine-dependent remodeling of neuronal RNP granules was significantly blocked upon Kir2.1-mediated hyperpolarization of MB neurons, a process that should inhibit evoked responses but not GPCR-mediated signaling. The identity of the TyrR-expressing neurons relaying Tyramine signal remains to be investigated. Previous experiments based on a knock-in line in which the *TyrR* locus has been deleted and replaced by Gal4 showed that this receptor is expressed in a restricted number of brain neurons, some of them (e.g. TyrIPS neurons) projecting in the vicinity of MBs and thus representing potential candidates (*Huang et al., 2016*). In the future, it will also be interesting to determine in which physiological or behavioral contexts the neuronal circuit activated by Tyramine regulates the translation of RNP granule-associated RNAs, and whether this regulation might be local. Although the physiological roles of Tyramine have so far been poorly characterized, this conserved bioamine has been involved in the regulation of both metabolic and behavior traits (*Lange, 2009*; *Ohta and Ozoe, 2014*; *Ma et al., 2016*; *Schützler et al., 2019*; *Roeder, 2020*). Potentially relevant to the known functions of MB neurons, Tyramine was in particular shown to regulate the avoidance behavior of flies exposed to repulsive olfactory cues (*Kutsukake et al., 2000*) and to dampen male courtship drive (*Huang et al., 2016*). Whether these responses involve post-transcriptional regulatory mechanisms such as those described in the context of long-term olfactory habituation in flies (*Bakthavachalu et al., 2018*) however remains to be investigated. Thus, testing whether Tyramine, similar to its derivative Octopamine (*Burke et al., 2012*; *Wu et al., 2013*), modulates learning and memory processes known to depend on RNA regulation represents another interesting line of future research.

# Materials and methods

## Key resources table

| Reagent type (species) or resource | Designation | Source or reference | Identifiers | Additional information |
|---|---|---|---|---|
| Genetic reagent (*D. melanogaster*) | w1118 | BDSC | RRID:BDSC_3605 | |
| Genetic reagent (*D. melanogaster*) | Tubulin-Gal80ts;; OK107-Gal4 | Besse team, IBV, Nice, FR | | This stock was obtained *via* genetic crosses so as to establish a stock carrying both the P{tubP GAL80ts} and the OK107 Gal4 driver. |

*Continued on next page*

*Continued*

| Reagent type (species) or resource | Designation | Source or reference | Identifiers | Additional information |
|---|---|---|---|---|
| Genetic reagent (*D. melanogaster*) | Canton-S | PMID:23083740 | | Dr K. Keleman (HHMI/ Janelia Farm, USA) |
| Genetic reagent (*D. melanogaster*) | MB247:homer:GCamp3.0 | PMID:24065891 | | |
| Genetic reagent (*D. melanogaster*) | TyrRGal4 | PMID:27498566 | | |
| Genetic reagent (*D. melanogaster*) | G080-GFP-Imp | PMID:24656828 | | Dr L. Cooley (Yale University, USA) |
| Genetic reagent (*D. melanogaster*) | VT44966-Gal4 | PMID:29322941 | RRID:FlyBase_ FBst0488404 | Dr K. Keleman (HHMI/ Janelia Farm, USA) |
| Genetic reagent (*D. melanogaster*) | YFP-CamkII | PMID:25294944 | RRID:DGGR_115127 | |
| Genetic reagent (*D. melanogaster*) | UAS-EGFP-kir2,1 | PMID:11343651 | RRID:BDSC_6595 | |
| Genetic reagent (*D. melanogaster*) | UAS-CamkII-ala | PMID:8384859 | RRID:BDSC_29666 | |
| Genetic reagent (*D. melanogaster*) | UASp-EGFP-*SV40*-3'UTR- | this study (Besse team, IBV, Nice, FR) | | Expresses EGFP coding sequence upstream of *SV40 3'UTR* under UAS control, insertion on chromosome III. For further information, see "*Generation of Drosophila lines*". |
| Genetic reagent (*D. melanogaster*) | UASp-EGFP-*profilin*-3'UTR | this study (Besse team, IBV, Nice, FR) | | Expresses EGFP coding sequence upstream of *profilin 3'UTR* under UAS control, insertions on chromosome II or III. For further information, see "*Generation of Drosophila lines*". |
| Genetic reagent (*D. melanogaster*) | UASp-EGFP-*cofilin*-3'UTR | this study (Besse team, IBV, Nice, FR) | | Expresses EGFP-coding sequence upstream of *cofilin 3'UTR* under UAS control, insertion on chromosome III. For further information, see "*Generation of Drosophila lines*". |
| Genetic reagent (*D. melanogaster*) | UASp-EGFP-*camk2*-3'UTR | this study (Besse team, IBV, Nice, FR) | | Expresses EGFP-coding sequence upstrream of *camk2 3'UTR* under UAS control, insertion on chromosome II. For further information, see "*Generation of Drosophila lines*". |
| Genetic reagent (*D. melanogaster*) | Me31B-GFP | this study (Besse team, IBV, Nice, FR ; Nakamura team, Kumamoto University, Kumamoto, Japan) | | Knock-in line generated using the CRISPR-Cas9 technology. The GFP tag is C-terminal. For further information, see "*Generation of Drosophila lines*". |

*Continued on next page*

Continued

| Reagent type (species) or resource | Designation | Source or reference | Identifiers | Additional information |
|---|---|---|---|---|
| Genetic reagent (*D. melanogaster*) | Me31B-mTomato | this study (Besse team, IBV, Nice, FR ; Nakamura team, Kumamoto University, Kumamoto, Japan) | | Knock-in line generated using the CRISPR-Cas9 technology. The mTomato tag is C-terminal. For further information, see "*Generation of Drosophila lines*". |
| Genetic reagent (*D. melanogaster*) | UAS-SunTag-*profilin* | this study (Besse team, IBV, Nice, FR ; Lagha team, IGMM, Montpellier, FR) | | Expresses a SunTagged Profilin (isoform RB) under UAS-control, insertion on chromosome II. For further information, see "*Generation of Drosophila lines*". |
| Genetic reagent (*D. melanogaster*) | UASp-scFv-GFP-NLS | this study (Besse team, IBV, Nice, FR ; Lagha team, IGMM, Montpellier, FR) | | Expresssd a scFvGFP under UAS-control, insertion on chromosome III. For further information, see "*Generation of Drosophila lines*". |
| Genetic reagent (*D. melanogaster*) | G080-GFP-Imp- RXXA mutant | This study (Besse team, IBV, Nice, FR) | | Mutant line generated using the CRISPR-Cas9 gene-editing technology. The four potential CamkII consensus sites RXXS/T are mutated into RXXA. For further information, see "*Generation of Drosophila lines*". |
| Antibody | Anti-Imp (Rabbit polyclonal) | PMID:24656828 | | IF (1:1000) |
| Antibody | Anti-Imp (Rat polyclonal) | PMID:24656828 | | IF (1:1000) |
| Antibody | anti-Me31B (Rabbit polyclonal) | PMID:28388438 | | Dr C. Lim (School of Life Sciences, Korea) IF (1:3000) and WB (1:5000) |
| Antibody | anti-Me31B (Mouse monoclonal) | PMID:11546740 | RRID:AB_2568986 | IF (1:3000) |
| Antibody | anti-pCamkII (rabbit polyclonal) | Santa Cruz Biotechnology | Cat# sc-12886-R RRID:AB_2067915 | IF (1:1000) |
| Antibody | anti-GFP (Chicken polyclonal) | Abcam | Cat# ab13970 RRID:AB_300798 | IF (1:1000) |
| Antibody | anti-GFP (Rabbit polyclonal) | Torrey Pines Biolabs | Cat# TP401 071519 RRID:AB_10013661 | WB (1:2500) |
| Antibody | anti-FLAG (Mouse monoclonal) | Sigma-Aldrich | Cat# F1804, RRID:AB_262044 | WB (1:2500) |
| Antibody | anti-Tubulin (Mouse monoclonal) | Sigma-Aldrich | Cat# T9026, RRID:AB_477593 | WB (1:5000) |

### *Drosophila* lines and genetics

Unless otherwise specified, flies were raised on standard media at 25°C and both males and females were dissected 9–14 days post-eclosion. The UAS-EGFP:kir2.1 and UAS-EGFP-3'UTR constructs were expressed under the control of tubulin-Gal80ts;OK107-Gal4. Flies were kept for 6 days after hatching at 21°C and then switched for 3 days at 29°C to trigger transgene expression. The CamkII inhibitory peptide ala construct was also expressed under the control of tubulin-Gal80ts;OK107-Gal4. Flies were kept throughout development at 21°C, switched to 29°C upon hatching, and dissected after 9–10 days. The SunTag-*profilin* and the ScFv-GFP constructs were expressed under the control of the VT44966-Gal4. The following fly stocks were used: w[1118]; tubulin-Gal80ts;;OK107-Gal4, CantonS (gift from Krystyna Keleman), MB247:homer:GCamP3.0 (*Pech et al., 2013*), TyrR[Gal4] mutants (*Huang et al., 2016*), G080-GFP-Imp (gift from L. Cooley, described in *Medioni et al., 2014*), VT44966-Gal4 (VDRC stock center), YFP-CamkII (CaMKII[CPTI000944], DGRC #115127), UAS-EGFP-kir2.1 (BDSC, #6595), and UAS-CamkII-ala (BDSC, #29666).

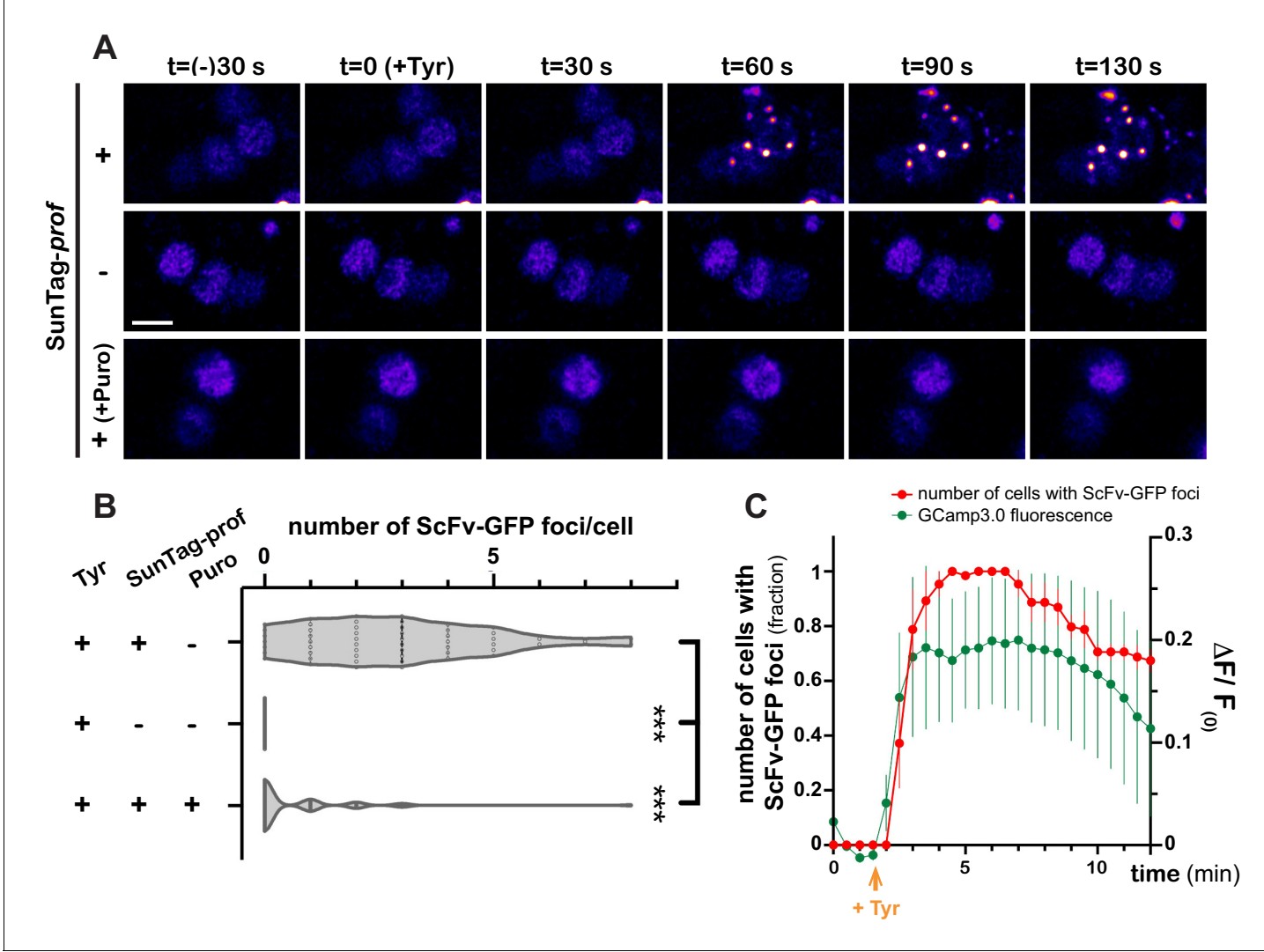

**Figure 6.** Dynamics of *profilin* translation upon Tyramine treatment. (**A**) Image sequences extracted from movies recording the distribution of SunTag-tagged Profilin peptides in brain explants. ScFv-GFP-NLS was expressed in MB γ neurons with (upper and lower panels) or without (middle panel) *SunTag-profilin* mRNAs. Images were recorded every 30 s and Tyramine was added at t = 0. The translation inhibitor puromycin (puro) was added prior to Tyramine (lower panel). Complete genotype: UAS-*SunTag-profilin*/+; UASp-ScFv-GFP-NLS/VT44966-Gal4. Scale bar: 3 μm. (**B**) Distributions of the number of Tyramine-induced ScFv-GFP-positive cytoplasmic foci observed per cell in the presence (+) or absence (−) of *SunTag-profilin* transcripts. Puromycin (puro) significantly inhibited the formation of ScFv-GFP-positive foci. ***, p<0.001 (Kruskal–Wallis test with Dunn's post-tests on individual data points). At least 45 cells and five movies were analyzed per condition. (**C**) Normalized number of cells with ScFv-GFP cytoplasmic foci (red) observed for each time point in brain explants treated with 10 mM Tyramine. Numbers of cells are normalized so that one represents for each movie the maximal number of cells with ScFv-GFP foci (n = 5 movies). Fluorescence intensity of GCamp3.0 is shown for comparison (green; see *Figure 3F*). Tyramine was added at t = 2 min (orange arrow). Error bars correspond to S.E.M. For the list of values used to generate the graphs shown in B, C see *Figure 6—source data 1*.

The online version of this article includes the following source data and figure supplement(s) for figure 6:

**Source data 1.** Numerical data to support graphs in *Figure 6B-C*.
**Figure supplement 1.** Model of Tyramine signaling and its impact on MB neuronal ribonucleoprotein (RNP) granules.

## Generation of *Drosophila* lines

The UASp-EGFP-3'UTR constructs were generated by LR recombination using pENTR:D/TOPO donor plasmids containing 3'UTR sequences and a UASp-EGFP-W destination vector. The UASp-EGFP-W plasmid was obtained by ligating into a KpnI-PstI blunted doubled-digested pPW plasmid, a KpnII-SacII insert obtained after subcloning into pBluescript (KS) a KpnI-EGFP-XhoI and a SpeI-

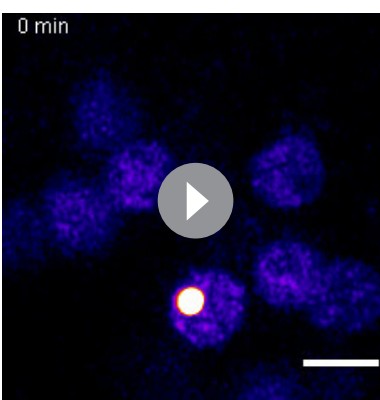

**Video 7.** Tyramine induces the assembly of SunTag-*profilin* foci. Real-time imaging of MB γ cell bodies co-expressing *SunTag-profilin* RNAs and ScFv-GFP-NLS fusions under the control of VT44966-Gal4. Note that VT44966-Gal4 is expressed at high level only in a subset of MB γ neurons. Some cells initially contained a big cluster of ScFv-GFP fusions; these cells were excluded from the analysis. Signal intensities are displayed using the 'Fire' LUT of ImageJ. Images were acquired every 30 s for 30 min, from intact adult brain explants. Tyramine was added at t = 2 min to reach a 10 mM final concentration. Scale bar: 3 μm.
https://elifesciences.org/articles/65742#video7

Gateway Cassette-SacII fragments. The *SV40* 3'UTR, *profilin* 3'UTR, *cofilin* 3'UTR and *camkII* 3'UTR sequences were PCR-amplified using the following primers: SV40_up (5'-CACCTAGAGGA TCTTTGTTGAAGG-3') and SV40_low (5'-GA TCCAGACATGATAAGATAC-3'); chic_up (5'-CACCGAGAATAGATCAACAC-3') and chic_low (5'-CGTGTGGATTTATGTACG-3'); cof_up (5'-CACCGACCGCCAATAAACTG-3') and cof_low (5'-TTGGTCAAGTTAAATATTTCATTCT-3'); camkII_up (5'-CACCACATTCGGATTTTATAC-3') and camkII_low (5'-TTATTATTATCTTTAAAAATTC-3').

The Me31B-GFP and Me31B-mTomato knock-in lines were generated using the CRISPR/Cas9 technology, as described in *Kina et al., 2019*. Briefly, *me31B* gRNA sequence was cloned into the pDCC6 plasmid using the following primers: me31B_sgRNA1F (5'-CTTCGAATAATTC TGCGAACGAGG-3') and me31B_sgRNA1R (5'-AAACCCTCGTTCGCAGAATTATTC-3'). To generate the GFP targeting vector, 5' and 3' homology arms were amplified by PCR using the following primers: me31B (4705–4685)+GFP (5'-GCCCTTGCTCACCATTTTGCTAACGTTGCCC TCCTC-3'); me31B (5'−4706–4726)+GFP (GAC GAGCTGTACAAGTAAAACGGATATGCCCTGTG T-3'); me31B (5'−3652–3671)+pBS (GGGAA CAAAAGCTGGATCCGGGTAATGGTCACAAC-3'); me31B (5'−5674–5655)+pBS (TATAGGGCGAATTGGACGATTCCCGATAATGCCAC-3'). These PCR products, the GFP coding sequence, and the *KpnI-SacI*-digested pBluescript SK plasmid were assembled into a sealed plasmid through Gibson assembly. A similar strategy was used for the Me31B-mTomato line.

The UAS-SunTag-*profilin* plasmid was generated using the NEBuilder HiFi DNA Assembly Master Mix (NEB#E2621), through two successive Gibson assembly reactions. In the first one, UASt and *profilin* 5'UTR fragments were assembled into the *twi_suntag_MS2 plasmid* backbone (*Dufourt et al., 2021*). The following primers were used for the PCR amplification of inserted fragments: UASt (1102_UASt_fwd 5'-tcgtcttcaagaattcgtttTGCTAGCGGATCCAAGCTTG-3', 1103_UASt_rev 5'-ttactttcaaTTCCCTATTCAGAGTTCTCTTCTTG-3'); *profilin* 5'UTR (1104_5'UTR_prof_fwd 5'-gaatagggaaTTGAAAGTAAGTTACCCCAAG-3', 1119_5'UTR_prof_rev 5'-gctgccgctaagcttggtCATacGGTGCTTTGTTTGTCGTG-3'). In the second one, *profilin* coding (cDNA) and 3'UTR sequences were assembled into the previously generated vector. The following primers were used for the PCR amplification of inserted fragments: *profilin* CDS (1133_profilin_fwd 5'-aaaaagggcagcgatatcaccggtagctggcaagat-tatgtg-3', 1134_profilin_rev 5'-atctattctcctagtacccgcaagtaatc-3'), *profilin* 3'UTR (1135_prof_utr_fwd 5'-cgggtactaggagaatagatcaacacaaacac-3', 1136_prof_utr_rev 5'-ggcgagctcgaattcactagtcgtgtggatt-tatgtacg-3'). The 3'UTR of *profilin* (isoform RB) was amplified from a pUC57-simple vector designed by gene synthesis so as to contain a NotI restriction site and (GenScript Biotech). MS2 128x repetitions (*Dufourt et al., 2021*) were cloned between FRT sequences and inserted into the NotI site. The final construct was injected in attP-VK00002 flies (BDSC #9723) using PhiC31 targeted insertion (BestGene, Inc). Note that the ms2 sequences were excised from the transgenic flies used in this study.

The UASp_scFvGFP_NLS plasmid was generated through Gibson assembly, by inserting the 10× UAS and p-transposase promoter sequences from pVALIUM22 into Not1/Xho1-digested pNo-sPE_scFvGFP_NLS plasmid (*Dufourt et al., 2021*) using NEBuilder HiFi DNA Assembly Master Mix. Fragments were amplified using the following primers: 965scfvgfpF (5'-ggccagatccaggtcg-cagcggccgcGCGGCCGCATAACTTCGTATAATG-3'); 966scfv (5'-cggggcccatCTCGAGTGA

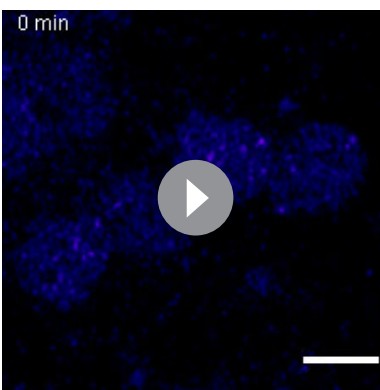

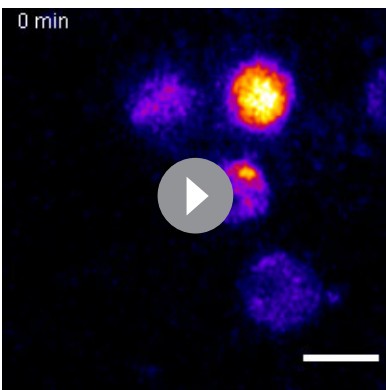

**Video 8.** ScFv-GFP-positive foci are not observed in the absence of SunTag-*profilin* transcripts. Real-time imaging of MB γ cell bodies expressing ScFv-GFP-NLS fusions under the control of VT44966-Gal4. Note that VT44966-Gal4 is expressed at high level only in a subset of MB γ neurons. Signal intensities are displayed using the 'Fire' LUT of ImageJ. Images were acquired every 30 s for 12 min, from intact adult brain explants. Tyramine was added at t = 2 min to reach a 10 mM final concentration. Scale bar: 3 μm.
https://elifesciences.org/articles/65742#video8

**Video 9.** Puromycin disrupts the assembly of SunTag-*profilin* foci. Real-time imaging of MB γ cell bodies co-expressing *SunTag-profilin* RNAs and ScFv-GFP-NLS fusions under the control of VT44966-Gal4. Note that VT44966-Gal4 is expressed at high level only in a subset of MB γ neurons. Signal intensities are displayed using the 'Fire' LUT of ImageJ. Images were acquired every 30 s for 12 min, from intact adult brain explants. Brain explants were incubated with 250 μM Puromycin for 15 min before addition of Tyramine. Tyramine was added 2 min after the movie starts, to reach a 10 mM final concentration. Scale bar: 3 μm.
https://elifesciences.org/articles/65742#video9

TCCCCGGGC-3'); 967uasp (5'-atcactcgagATGGGCCCCGACATCGTG-3'); 968scfvgfp (5'-cattgtgtgagttaaagttgtactcgagTTTGTGTCCAA-GAATGTTTCCATCTTCTTTAAAATC-3').

The line expressing GFP-Imp proteins with mutated CamkII consensus sites from the endogenous locus was generated through CRISPR/Cas9 gene editing, following a two-step procedure. First, an *imp*-RMCE line in which the upstream GFP exon of the G080 line was preserved, but most of the *imp* locus deleted and replaced by a cassette containing the 3xP3-RFP selection marker flanked by inverted attP sites was generated by homology-dependent repair, using two gRNAs (upstream: ACATTGCATTGCAGCTGAGTTGG and downstream: GCGAGCTCACAACAGTAAGGAGG) and a dsDNA donor plasmid with 1 kb long-homology arms (chrX:1078009–10799070 and 10797081–10798308; Wellgenetics). Second, a donor pBS-KS-attB1-2 plasmid in which the four RXXS/T consensus sites of Imp were mutated into RXXA was generated through Gibson assembly and integrated through cassette exchange (RMCE) in the *imp*-RMCE line. Individuals with a correctly oriented cassette were selected by PCR and their genomic *imp* locus sequenced.

## Ex vivo treatments of *Drosophila* brains

Brains of 9–12-day-old flies were dissected in cold Haemolymph-Like saline solution 3 (HL3) (NaCl 70 mM, KCl 5 mM, MgCl$_2$ 4 mM, trehalose 5 mM, sucrose 115 mM, HEPES 5 mM, NaHCO$_3$ 10 mM, pH 7.2–7.3) and transferred into Nunc Lab-Tek II Chamber Slide (Thermofisher, #154526) with either 500 μL of HL3 or HL3 supplemented with neurotransmitter for 30 min at 25°C. Brains were protected from light during incubations. Neurotransmitters were used at the following final concentrations: Acetylcholine (Sigma, # A6625): 10 mM; Tyramine (Sigma, # T2879): 10 mM; Dopamine (Sigma, # H8502): 10 mM; Octopamine (Sigma, # O0250): 10 mM. For treatment with the translational inhibitor anisomycin (sigma, #A9789), anisomycin was added 20 min prior to Tyramine at a final concentration of 40 μM and maintained throughout Tyramine treatment. After treatment, brains were collected, fixed with 4% formaldehyde in HL3 for 25 min, washed thrice with phosphate-buffered saline supplemented with 0.5% Triton-X (PBT) and either directly mounted in vectashield (Vector Laboratories) to image endogenous fluorescence or further immuno-stained.

## Immunostainings

After fixation and washes in PBS/Triton-X (PBT) 0.5%, brains were blocked overnight in PBS/Triton-X (PBT) 0.5% supplemented with Bovine Serum Albumin (BSA) 1% and then incubated with the following antibodies: α-Imp (rabbit, 1:1000; *Medioni et al., 2014*), α-Imp (rat, 1:1000; *Medioni et al., 2014*), α-Me31B (rabbit, 1:3000; *Lee et al., 2017*), α-Me31B (mouse, 1:3000; *Nakamura et al., 2001*) α-pCamkII (rabbit, 1:1000; Santa Cruz Biotechnology, sc-12886-R), α-GFP (chicken, 1:1000; Abcam, #ab13970). After incubation with primary antibodies, brains were washed three times with PBT 0.5% and incubated overnight with secondary antibodies. The following secondary antibodies were used in this study: Goat anti-rat AF568 (Thermofisher, A-11077), Goat anti-rat AF647 (Thermofisher, A-21247) Donkey anti-rabbit AF568 (Thermofisher, A-10042), Donkey anti-rabbit AF647 (Thermofisher, A-31573), Donkey anti-mouse AF488 (Thermofisher, A-21202), Donkey anti-mouse AF568 (Thermofisher, A-10037), Donkey anti-mouse AF647 (Thermofisher, A-31571), and Goat-anti-chicken AF488 (Thermofisher, A-11039). DAPI was used at 5 µg/mL and incubated for 5 min after secondary antibody incubation. Brains were washed in PBT 0.5% three times following secondary antibody incubation and were then mounted in vectashield (Vector Laboratories).

## smFISH

*Drosophila* brains were dissected in cold RNase-free HL3 and treated for 10 min with 10 mM Tyramine. Samples were then fixed for 1 hr in 4% formaldehyde in PBS and dehydrated overnight in ethanol 70%. Brains were then briefly rinsed in wash buffer (10% formamide in 2× SSC) before overnight incubation at 45°C in Hybridization Buffer (100 mg/mL dextran sulfate, 10% formamide in 2× SSC) supplemented with Quasar 570-labeled Stellaris Probes at a final concentration of 0.25 µM. Brains were then washed twice in pre-warmed wash buffer, stained with DAPI 5 µg/mL, briefly washed in 2× SSC and mounted in vectashield (Vector Laboratories). Sequences of the probe sets designed to hybridize to *profilin* and *gfp* were as follows (from 5' to 3'): *profilin*: ccgcaacaccgacgattt, cacacgaaattggcaggg, tcgcactttcgtttcggg, ttgctttaccgcacggcg, gatctggatatggatcgc, gggtgcggattaagttga, catggtgctttgtttgtc, gtccacataatcttgcca, ctgcgaggccaggagttg, gatgcacgccttggtcac, ccaaatgttgccgtcgtg, tcacctcaaagccactgg, gtttggagagctcctctt, ctggtcaaagccgctgat, gttgctggtgagaccgtc, aaatgtaccgctggccgg, gcggtctgtgccggaaag, ttcatgcagtgcactccg, acgatcacggcttgtgtt, cgggatcctcgtagatgg, tctctaccacggaagcgg, ctattctcctagtacccg, tcatttacggttcgtctct, tggtttttcttttcccat, gcaaattctttcttggcc, tcctctgctacacacaaa, gcatttttactcgatcca and *gfp*: tcctcgcccttgctcaccat, atgggcaccccggtgaa, gtcgccgtccagctcgacca, cgctgaacttgtggccgtttt, tcgccctcgccctcgccgga, tcgccctcgccctcgccgga, tcgccctcgccctcgccgga, tcgccctcgccctcgccgga, tcgccctcgccctcgccgga, tcgccctcgccctcgccgga, tcgccctcgccctcgccgga, ggtcagcttgccgtaggtgg, ggccagggcacgggcagctt, taggtcagggtggtcacgag, tagcggctgaagcactgcac, gtgctgcttcatgtggtcgg, gcatggcggacttgaagaag, cgctcctggacgtagccttc, gtcgtccttgaagaagatgg, tcggcgcgggtcttgtagtt, ggtgtcgccctcgaacttca, ttcagctcgatgcggttcac, gtcctccttgaagtcgatgc, agcttgtgccccaggatgtt, gtggctgttgtagttgtact, ttgtcggccatgatatagac, caccttgatgccgttcttct, atgttgtggcggatcttgaa, gagctgcacgctgccgtcct, tgttctgctggtagtggtcg, agcacggggccgtcgccgat, caggtagtggttgtcgggca, ttgctcagggcggactgggt, atcgcgcttctcgttggggt, cgaactccagcaggaccatg, agagtgatcccggcggcggt, cttgtacagctcgtccatgc.

## Image acquisition

Fixed samples were imaged using a Plan Apo 63X NA 1.4 oil objective, on a Zeiss LSM880 inverted confocal microscope equipped with an Airy Scan module. Images were acquired with a pixel size of 0.043 µm and were processed with the automatic Airy Scan processing mode (strength 6.0). Note that MB γ neurons were located based on their position within MBs (estimated by DAPI) and the differential expression of Imp in MB neuron sub-types (*Medioni et al., 2014*).

## Live imaging

Brains of 10–15-day-old flies were dissected in HL3 at room temperature, and then transferred and mounted on a four chambered 35 mm dish with 20 mm bottom well (IBL, #D35C4-20-0-N) poly-lysinated before use (3 hr incubation at RT or overnight at 4°C). Once correctly oriented, brains were stabilized on the plate using low melting agarose (NuSieve GTG Agarose #50080, 0.07%, dissolved in HL3). Live imaging of RNP granule remodeling and time-course calcium imaging was performed using a 40× NA 1.1 water objective, on a Zeiss LSM880 inverted confocal microscope equipped with

an Airy Scan module. Images were acquired every 30 s for 32 min. Tyramine (dissolved in HL3) or HL3 were added as a single drop 2 min after the start of imaging. To inhibit translation, puromycin (sigma, #P8833) was applied at a final concentration of 250 µM and incubated for 15 min prior to addition of Tyramine.

For Tyramine dose–response experiments, MB247-homer::GCamp3.0 brains were mounted on 35 × 10 mm petri dishes (Nunc, #153066) coated with poly-lysine and covered with HL3. MB calyx regions were then imaged with an upright Leica DM6000 TCS SP5 confocal microscope equipped with a HCX APOL 40X water (0.8 NA) objective. Two images of MB calyces were taken for each brain: one at t = 0 (before addition of Tyramine) and one at t = 30 min.

## Image analysis
### RNP granule and RNA detection
For RNP quantifications, 114.9 µm² ROIs containing six to seven cells were cropped from single z slices (two independent ROIs per brain), treated with a Gaussian Blur filter, resized by a factor two using the Image Pyramid plugin and converted from 32-bit to 16-bit images, all in ImageJ. Granules were detected and quantified using the Small Particle Detection (SPaDe) algorithm available under the following link : https://hal.inria.fr/hal-01867805/document. Minimal granule size was set to four pixels and threshold defined so as to optimize measured F1 scores (as described in *De Graeve et al., 2019*). F1 scores were calculated by comparing the spatial coordinates of manually annotated RNP granules with the binary masks generated by SPaDe with different thresholds. Threshold used for detection of Imp and Me31B-positive granules was hence set to 0.6234. In experiments where Imp antibody staining was used, RNP granules of a size smaller than 13 pixels were excluded from analysis. A similar procedure was used for detection of smFISH RNA spots, using 0.3434 as SPaDe threshold.

### Me31B partition coefficient measurements
Me31B partition coefficients were defined as the ratio between the maximal intensities of individual RNP granules and the average diffuse cytoplasmic signal. Maximal RNP granule intensities were measured with ImageJ on the original raw images, using the masks generated by SPaDe. For cytoplasm measurements, masks were obtained using the following procedure: first, a Gaussian Blur filter (σ:8) was applied on images in which granule-containing pixels were previously blanked; second, the mean of the intensity signal was measured and used as lower limit for a thresholding interval (the upper limit being the maximum possible intensity value, 65000). Mean intensities were then measured on raw images using these masks.

To monitor RNP remodeling over time, two regions of 102.6 µm², each containing six to seven cells, were cropped per movie. For each time point, Me31B partition coefficients were calculated for each granule of the cropped regions as described previously, and mean partition coefficients calculated.

### Me31B and Imp protein levels
Protein signals in the whole cytoplasm were measured in living samples at t = 0 and t = 32 using two regions per movie of respectively 102.6 µm² for Me31B and 71.2 µm² for Imp. Individual t = 32 values were normalized to their corresponding t = 0 measurements.

### Intracellular calcium quantification
Real-time calcium imaging was quantified as follows: a 155 µm² region was selected in the MB calyx of each imaged hemisphere and average intensity measured for each time point. Data were plotted as F(t)-F(0)/F(0), where F(0) is the mean of the four intensity values obtained before Tyramine was added (t0 to t3), and F(t) represents the intensity value measured at time t.

For the dose-dependent c response to Tyramine, GCamp3 mean fluorescence intensity was measured for each hemisphere in ImageJ, on the t = 0 and t = 30 images using a 17.5 µm² ROI. Measures were normalized such that the value of each t0 was set to 1.

### ScFv-GFP puncta analysis

The number of ScFv-GFP-containing spots was counted manually from movies. Cells containing at t = 0 a big cluster of ScFv-GFP fusions were excluded from the analysis.

### Translational reporter expression

For the 3'UTR reporter experiments, average GFP intensity was measured with ImageJ on maximum intensity projections, within cell body regions of 173.6 μm². Two ROIs were considered per brain.

### Colocalization analysis

To quantify the number of *profilin* RNA contained in RNP granules, binary masks were first generated by SPaDe from the RNP granule channel (GFP-Imp or Me31B-mTomato) and from the smFISH channel (*GFP-profilin* or *profilin*). Co-localization was then assessed using the ImageJ JACoP plugin (https://imagej.nih.gov/ij/plugins/track/jacop.html) and the object-based method (**Bolte and Cordelières, 2006**), where RNA spots were defined by their center of mass and RNP granules considered as particles, respectively. The fraction of RNP spots contained in granules was calculated for each condition.

## Immunoprecipitation–mass spectrometry

201Y-Gal4, UAS-GFP-Imp and 201-Gal4, UAS-GFP flies were amplified at 25°C in bottles. 3–5-day-old flies were collected and immediately frozen. Heads were collected at 4°C using two prechilled sieves of different mesh sizes (630 μm on top and 400 μm at the bottom) and crushed into powder using prechilled mortar and pestle. The head powder (3 g for each experiment) was then transferred to a prechilled 15 mL glass Dounce Tissue Grinder and homogenized in 10 mL DXB buffer (25 mM HEPES), pH 6.8, 50 mM KCl, 1 mM MgCl₂, 1 mM dithiothreitol (DTT), 250 mM sucrose, 1/100 Halt Protease, and Phosphatase Inhibitor Cocktail (# Thermo Scientific). The homogenate was cleared by two consecutive centrifugations (10,000 rpm for 10 min at 4°C). 250 μL of GFP-Trap_A beads (ChromoTek, Germany) were then added to the cleared lysate in a 15 mL falcon tube and incubated on a rotator for 1.5 hr at 4°C. Beads were pelleted by mild centrifugation (2000 rpm for 2 min at 4°C), washed twice with DXB buffer and once with DXB buffer + 0.1% NP 40. Proteins bound to the beads were eluted by addition of 1.6 mL of 0.2M glycine, pH 2.5 and incubation for 10 min on a rotator. Eluates were then collected and neutralized with 400 μL of 1M Tris-HCl, pH 8, before being concentrated on an Amicon Ultra-2 mL centrifugal filter, MWCO 10 kDa (Merck Millipore, USA) to obtain a final volume of 50 μL that was loaded on a polyacrylamide gel. Coomassie-stained gels were cut, trypsin-digested, and further processed through LC-MS/MS by the EMBL proteomic core facility. Peptides were searched using Mascot (search engine, v.2.2.07) against the Uniprot_*Drosophila melanogaster* database, and then uploaded and analyzed using the Scaffold three software. Proteins fulfilling the following requirements were considered as Imp interactors: (i) represented by at least 10 peptides, (ii) represented with at least twice more peptides in the GFP-Imp pull-downs than in the GFP control IPs, and (iii) meeting the previous requirements for the two independent replicates performed.

## Subcloning of CamkII coding region

CamkII sequence was amplified by PCR from cDNA obtained from flies carrying a UAS-CamkII.R3 construct (BSC #29662), using the following primers: CaMKII_Gtw_for (5'-CACCATGGCCGCAC-CAGCAGC-3') and CaMKII_Gtw_rev (5'-TATTTTTGGGGTATAAAATCG-3'). The obtained sequence was subcloned into pENTR-D/TOPO vector (Life Technologies), fully sequenced, and then LR-recombined into the pAFW vector.

## Immunoprecipitation experiments

*Drosophila* S2R+ cells were plated in 6-well plates at a density of $5.10^6$ cells/well and incubated for 24 hr at 25°C. Cells were then transfected with 600 ng of plasmids using Effecten transfection reagents (Qiagen, 301425) and supplemented after 12 hr with Schneider's Insect Medium containing fetal bovine serum (10%) and penicillin/streptomycin (1%). After 3 days of expression at 25°C, cells transfected with either pAGW + pAFlag-CamkII or pAGFP-Imp + pAFlag-CamkII plasmids were harvested and lysed in DXB buffer (HEPES pH 6.8, 2.5 mM; KCl, 50 mM; MgCl₂, 1 mM; DTT, 1 mM;

sucrose, 250 mM; NP-40, 0.05%) supplemented with Halt Protease Inhibitor Cocktail 1:100 (Thermofisher, #78429). This initial lysate was used as input material and then split into two halves, one treated (+ RNase) with 10 mg/µL RNase A/T1 Cocktail (ThermoFisher scientific, #EN0551) and the other not (− RNase). Both cell lysates were subsequently incubated with ChromoTek GFP-Trap beads (ChromoTek, gt-10, #70112001A) for 2 hr at 4°C. Three washes were then performed with DBX supplemented with Halt Protease Inhibitor Cocktail 1:100, and bound fractions eluted in SDS 2× buffer by incubating the beads at 95°C for 5 min.

## Western blots

Fifteen brains were lysed in 40 µL RIPA Buffer (0.1% Na deoxycholate, 1% Triton-X, 0.1% SDS, NaCl 150 mM, Tris-HCl pH 7–7.5 50 mM) supplemented with Halt Protease Inhibitor Cocktail 1:100 (Thermofisher, #78429) and SDS 5× buffer. Lysates were agitated for 30 min at 4°C and then incubated for 5 min at 95°C for denaturation prior to loading.

Protein extracts were run on 10% polyacrylamide gels, blotted to PVDF membranes, and incubated first in blocking solution (PBS1×: milk 5%) and then overnight (4°C) with the following primary antibodies: rabbit anti-GFP (Torrey Pines, #TP401, 1:2,500), mouse anti-FLAG (Sigma, #F1804, 1:2,500), mouse anti-tubulin (Sigma, #T9026, 1:5,000), rabbit anti-Me31B (C. Lim, 1:5000). Membranes were washed three times with PBS:Tween-20 0.1% and then incubated for at least 2 hr at RT with the following secondary antibodies diluted in PBS:Tween-20 0.1%: goat anti-rabbit AF680 (Invitrogen, #A21076, 1:10,000), goat anti-mouse IRDye 800 (Invitrogen, #SA-10156, 1:10,000). The membranes were then washed three times with PBS:Tween-20 0.1%. The fluorescent signal was revealed using an Odyssey LI-COR system.

## Statistical analysis

Statistical details for each experiment can be found in the corresponding figure legends. SuperPlots were used to show individual data points with a color code referring to the original experimental set. Average values of each biological replicate are shown using the same color-code (*Lord et al., 2020*).

All statistical analyses were performed using GraphPad Prism 8.

## Acknowledgements

This study was supported by the CNRS, as well as ANR grants (ANR-15-CE12-0016 and ANR-20-CE16-0010-01) and a Fondation pour la Recherche Médicale grant (Equipe FRM; grant #DEQ20180339161) to FB. Part of this work was also supported by an ERC grant (SyncDev) to ML, a JSPS KAKENHI grant (#17H03686) to AN, and funding from the Joint Usage/Research Center for Developmental Medicine, IMEG, Kumamoto University. NF was supported by a fellowship from the LABEX SIGNALIFE program (#ANR − 11 − LABX − 0028–01) and JD was supported by a HFSP CDA grant to ML. We thank L Palin for excellent technical assistance, F De Graeve for help with granule analysis, and A Majumdar for discussion and advise. We thank the PRISM Imaging Facility for use of their microscopes and support, especially B Monterroso for his help with AiryScan imaging. We thank the EMBL Proteomics Core Facility for performing the LC-MS/MS analysis. We are grateful to A Hübstenberger and C Medioni for critical reading of the manuscript. We are grateful to the Bloomington *Drosophila* Stock Center and the Developmental Studies Hybridoma Bank for reagents and to the lab of E Bertrand for sharing the 32X SunTag plasmid.

## Additional information

### Funding

| Funder | Grant reference number | Author |
|---|---|---|
| Agence Nationale de la Recherche | ANR-15-CE12-0016 | Florence Besse |
| Agence Nationale de la Recherche | ANR-20-CE16-0010-01 | Florence Besse |

| Fondation pour la Recherche Médicale | DEQ20180339161 | Florence Besse |
| H2020 European Research Council | SyncDev | Mounia Lagha |
| Agence Nationale de la Recherche | ANR-11-LABX-0028-01 | Florence Besse |
| Human Frontier Science Program | CDA | Mounia Lagha |
| Japan Society for the Promotion of Science | KAKENHI-17H03686 | Akira Nakamura |

The funders had no role in study design, data collection and interpretation, or the decision to submit the work for publication.

## Author contributions

Nadia Formicola, Conceptualization, Formal analysis, Validation, Investigation, Visualization, Methodology, Writing - review and editing; Marjorie Heim, Anne-Sophie Lancelot, Investigation, Methodology; Jérémy Dufourt, Conceptualization, Investigation, Methodology, Writing - review and editing; Akira Nakamura, Investigation, Methodology, Writing - review and editing; Mounia Lagha, Conceptualization, Supervision, Funding acquisition, Methodology, Project administration, Writing - review and editing; Florence Besse, Conceptualization, Formal analysis, Supervision, Funding acquisition, Visualization, Methodology, Writing - original draft, Project administration, Writing - review and editing

## Author ORCIDs

Nadia Formicola (iD) https://orcid.org/0000-0002-2039-1323
Marjorie Heim (iD) https://orcid.org/0000-0002-7858-4822
Jérémy Dufourt (iD) https://orcid.org/0000-0003-4043-7250
Akira Nakamura (iD) http://orcid.org/0000-0001-6506-9146
Mounia Lagha (iD) https://orcid.org/0000-0002-7082-1950
Florence Besse (iD) https://orcid.org/0000-0003-4672-1068

## Decision letter and Author response

Decision letter https://doi.org/10.7554/eLife.65742.sa1
Author response https://doi.org/10.7554/eLife.65742.sa2

# Additional files

## Supplementary files

• Supplementary file 1. Imp protein interactors identified through IP-MS. Number of peptides found for each protein in the GFP-Imp- or GFP- bound fractions are indicated for replicates 1 and 2.

• Transparent reporting form

## Data availability

All data generated or analysed during this study are included in the manuscript and supporting files.

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
