## [Decision Letter]

[Editors' note: this paper was reviewed by Review Commons.]

**Acceptance summary:**

In this interesting and timely manuscript, Formicola et al. address the question whether external stimuli affect RNP assembly and mRNA regulation. The authors convincingly show that the bioamine tyramine induces reversible remodelling of RNPs containing the RNA-binding proteins Imp and Me31b. This remodelling is dependent on neuronal signalling through the TyrR receptor and on the activity of CamkII. Finally, tyramine treatment resulted in the translational derepression of Imp associated mRNAs. In sum, the data presented here should be of immediate interest to several fields, including RNA biology, RNP granules and learning and memory.

**Decision letter after peer review:**

Thank you for submitting your work entitled "Tyramine induces dynamic RNP granule remodeling and translation activation in the *Drosophila* brain" for consideration by *eLife*. Your article, with proposed responses and additions to feedback from 3 peer reviewers has been reviewed by Mani Ramaswami as Reviewing Editor and K VijayRaghavan as the Senior Editor.

In this interesting and timely manuscript, Formicola et al. address the question whether external stimuli affect RNP assembly and mRNA regulation. The authors convincingly show that the bioamine tyramine induces reversible remodelling of RNPs containing the RNA-binding proteins Imp and Me31b. This remodelling is dependent on neuronal signalling through the TyrR receptor and on the activity of CamkII. Finally, tyramine treatment resulted in the translational derepression of Imp associated mRNAs. In sum, the data presented here should be of immediate interest to several fields, including RNA biology, RNP granules and learning and memory.

The manuscript has been improved but there are some remaining issues that need to be addressed, as outlined below:

1. The experiments presented do not seem to show that the axonal / synaptic mRNP granules are altered by tyramine. Instead they show tyramine regulation of somatic granules, which may be reasonable, easily assayable surrogates for synaptic granules. While this is acceptable, it is recommended that this experimental gap be clearly acknowledged, because granules in neuronal processes are most likely the ones important to regulate for memory-related processes.

2. Please consider if the language used to describe the association of Imp and CamKII is optimal. Particularly, whether terms such as 'partner' and 'interaction' to discuss this issue may be subject to misinterpretation. It could be better to state that these proteins are found in the same complex etc.

3. Consider including citations to what appear to be relevant previous work. In particular Barbee et al. 2006 (10.1016/j.neuron.2006.10.028) which is probably the first report of Imp on a class of neuronal granule; and Bakthavachalu et al. 2018 (10.1016/j.neuron.2018.04.032), which describes experiments that perhaps most definitively argue for a function for neuronal RNP granules (and by inference their regulation) in long-term memory.

---

## [Author Response]

Reviewer #1:In this interesting and timely manuscript, Formicola et al. address the question whether external stimuli affect RNP assembly and mRNA regulation. The authors convincingly show that the bioamine tyramine induces reversible remodelling of RNPs containing the RNA-binding proteins Imp and Me31b. This remodelling is dependent on neuronal signalling through the TyrR receptor and on the activity of CamkII. Finally, tyramine treatment resulted in the translational derepression of Imp associated mRNAs. In sum, the data presented here should be of great interest to several fields, e.g. RBP biology and learning and memory.Overall, the manuscript is well written and easy to follow. The figures are well designed and intuitive to understand. There is no issue at all about the data being reproducible. In general, the experiments are well designed to address the proposed questions and the data analysis and presentation is fitting. In the remainder, there are suggestions that will hopefully help the authors to improve their already strong story before publication.Significance: In sum, the data presented here should be of great interest to several fields, e.g. RBP biology and learning and memory. Overall, the manuscript is well written and easy to follow. The figures are well designed and intuitive to understand. In general, the experiments are well designed to address the proposed questions and the data analysis and presentation is fitting.Major Comments:In Figure S1D, the authors make the important point that decondensation is reversible for Imp granules. It would be interesting to see if Me31b is also reassembled into the same granules.

We agree and have performed new experiments where we compared the partition coefficient of Me31B observed after Tyramine treatment with that observed after subsequent recovery. As shown in the revised Figure 1—figure supplement 1, Me31B efficiently re-partitioned into granules after recovery, similarly to what was observed for Imp.

In Figure 3 it would be interesting to see the effects of TyrR-/- without tyramine.In Figure 4BC, the authors show the effects of the ala peptide under tyramine treatments. It would be interesting to see if the ala peptide triggers a phenotype without tyramine.

We thank the reviewer for the two related suggestions. We had analyzed the phenotype of TyrR-/- (Figure 3) and ala (Figure 4) in the absence of Tyramine treatment but, as no significant difference was observed with the +/+ controls, decided to not show the data for the sake of clarity. As suggested by the reviewer, and as these data may be useful to interpret the experiments, we have included these controls in the revised Figures 3D,E and Figures 4B,C.

On page 7, the authors argue that TyrR expressing neurons might relay to MB neurons. Which neurons could this be?

Although we don’t know the identity of these neurons, we now speculate in our revised discussion (page 12) about putative candidates.

Does RNP remodelling require both evoked responses and GPCR-mediated signaling?

Indeed, our results indicate that RNP remodeling require both evoked responses (Figure 1—figure supplement 1D) and GPCR-mediated signaling (Figure 3D,E). We added a sentence page 7 to clarify this point.

Again, it would be interesting to see the data for Kir2.1 without tyramine treatment.

We agree with the reviewer and have included in our revised Figure 1—figure supplement 1D the Kir2.1 without Tyramine condition that we had analyzed but not shown on the initial graph. No difference was observed with the control condition.

On page 8 and 11, the authors indicate a physical interaction between Imp and CamKII. The authors may want to rethink indicating a direct interaction here, as the data do not exclude other proteins acting as mediators.

We agree with the reviewer that the interaction we have uncovered may not be direct and have replaced the term “physical interaction” by “association” (pages 4 and 8) or “interaction” (page 11).

It would be interesting to see whether profilin mRNA changes its localization to Me31b granules upon tyramine treatment to determine the effects in correlation to translational derepression.

We thank the reviewer for this suggestion. To address this point, we performed smFISH experiments to detect *gfp-profilin* 3’UTR reporter RNA, or endogenous *profilin* transcripts, in brains expressing fluorescent granule markers. As shown in our revised Figure 5D and Figure 5—figure supplement 1A, a significant decrease in the number of *profilin* transcripts associated with granules was observed in both cases after 10 minutes of Tyramine treatment (*i.e.* at the time of translational derepression), indicating that the release of mRNAs from RNP granules represents an early step of granule remodeling that temporally matches with translation activation.

This would also be interesting in the context of the SunTag data, where the authors show that nascent peptides can be detected within one minute after tyramine treatment. However, in Figure 2 it is seen that condensation of ME31b occurs somewhat slower. This raises the questions:– if the decondensation kinetics of Imp are faster than those of Me31b,

To determine if Imp follows the same decondensation kinetics as Me31B, we treated GFP-Imp-expressing brain explants with Tyramine and imaged them in real-time for 30 minutes. As shown in our revised Figure 2—figure supplement 1, GFP-Imp appears to not decondense faster than Me31B.

– or if profilin mRNA quickly relocalizes,

To address this point, we have performed smFISH experiments 10 minutes after Tyramine treatment, *i.e.* at a time when translation is activated, but RNP component decondensation is only partial. As discussed above, our results (shown in Figure 5D and Figure 5—figure supplement 1A) indicate that granuleassociated *profilin* transcripts indeed quickly relocalize to the cytoplasm upon Tyramine treatment.

– or if the nascent SunTag signals colocalize with Imp/Me31b granules,

To perform this experiment, and image in real-time both SunTag signals and Imp/Me31B granules, we would need a chromosome expressing endogenous Imp (or Me31B) fused to a red fluorescent protein. Despite multiple attempts to CRISPR-tag Imp and Me31B with mcherry, mTomato or Apple (either Cterminally or N-terminally), we so far did not manage to obtain a line in which red fluorescent proteins do not ectopically aggregate in the context of live-imaging. Furthermore, for this experiment to be unambiguously interpreted, we also would need to image the RNA in addition to the SunTag signal and the Imp/Me31B signal, which is beyond current technical limits.

– or if only already cytoplasmic profilin is translated.

Although we cannot formally exclude this possibility, our new results indicate i- that the total amount of *profilin* transcripts is not modified by Tyramine treatment (revised Figure 5—figure supplement 1B), and ii- that the translational de-repression of *profilin* temporally matches with a release of *profilin* transcripts from RNP granules (Figure 5D and Figure 5—figure supplement 1A). These new results thus further suggest that the observed increase in translation reflects the de-repression of transcripts initially associated with granules.

Together, our new results have shown that the release of granule-associated transcripts occurs before quantifiable decondensation of both Imp and Me31B, with kinetics matching that of translation activation.

Minor Comments:On page 8, the authors indicate that the ala peptide was expressed in MB neurons. However, was the ala peptide not expressed globally (e.g. other brain regions)? If so the authors may want to discuss if this has relevance for their hypothesis, that other neurons modulate MB neurons via tyramine.

In these experiments, the ala peptide was conditionally and specifically expressed in adult MB neurons. We tried to make this clearer in the legend to Figure 4.

Figure 4C is falsely referenced on page 9. The authors probably wanted to reference Figure 4D, which is not referenced at any other point.

This is correct and we have changed 4C into 5C (4D has moved to 5C, see below).

Perhaps the authors might consider moving 4D to Figure 5.

As suggested by the reviewer, we have moved Figure 4D to the revised Figure 5 (5C).

Perhaps label the y-axis with GFP-profilin.

We thank the reviewer for this suggestion. We have now better labeled the y-axis in graphs 5B and 5C.

The authors should include example videos of the data collected for Figure 3F.

We have now added two videos illustrating the Ca^2+^ response of MB neurons in control (revised Video 5) and TyrR-/- (revised Video 6) brains treated with Tyramine.

It would be interesting to discuss why Me31b was not among the highly enriched targets of the mass spec data set.

This is an interesting point. Although RNA binding proteins enriched in Imp+ granules could be identified in our Mass-Spectrometry experiments (*e.g.* Heph, Pur-α), Me31B, as well as other granule components (*e.g.* Staufen, Tral), were not. We think this reflects the fact that our immuno-precipitation experiments did not specifically target the granule-associated pool of Imp but rather were probably enriched in cytoplasmic soluble complexes. We have now added a sentence to explain this hypothesis in the Results section (page 8).

On page 7, the authors argue that TyrR might be the main receptor involved in the observed decondensation of RNPs. The authors should shortly discuss the possible role other tyramine receptors might have, perhaps on page 11.

We thank the reviewer for this comment. Our conclusion that TyrR is the main receptor involved in RNP decondensation is based on experiments shown in Figure 3D,E, in which we demonstrate that genetic inactivation of *TyrR* nearly abolishes Tyramine-decondensation of RNPs. Although we cannot exclude that other receptors may be marginally implicated, we do believe that TyrR has a predominant function. To take into account the reviewer comment, we have clarified the identity of the other known Tyramine receptors in our revised discussion (page 12), and highlighted that these receptors are not significantly expressed in MBs.

If the authors are willing to share their unpublished data regarding the mutation of CamkII phosphorylation sites on Imp discussed on page 11, this would make an interesting and highly appreciated additional (supplemental) figure.

We thank the reviewer for this suggestion. Quantitative analysis of this experiment is now shown in Figure 4—figure supplement 1D. How the line expressing GFP fusions with mutated CamkII consensus sites has been generated is also now explained in the revised Materials and Method section.

Reviewer #2:Summary:In the impressive manuscript, Formicola et al. demonstrate tyramine-mediated decondensation of RNP granules in the intact *Drosophila* brain and show that this process is associated with translational activation of mRNAs that associate with these structures. It is also shown that this process is mainly driven by a single tyramine receptor and involves CamKII, which they find is in a complex with the RNAbinding protein Imp. Almost all conclusions are strongly supported by the data (see below for one exception), which come from an impressive array of technical approaches. The manuscript is well written and the data well controlled and presented.Significance:Whilst RNP granule dynamics have been studied in other contexts, there is very little information on how these processes are orchestrated in neurons, particularly within an intact animal. Therefore the study has significant novelty. The principle that a neuromodulator can impact of granule decondensation is a novel concept that is likely to interest researchers working on both behaviour and molecular cell biology of biological condensates. The impact would be greater if the authors were able to find a direct target for CamKII in this system but it appears that they have already tried hard to do so. The study would also be more impactful if a connection could be made between behaviour and RNP dynamics, but this is likely to be beyond the scope of this study (which already covers a lot of ground).Major comments:1. The authors conclude from the Kir2.1 overexpression experiments (Figure S1E) that tyramine-receptive neurons relay the signal to MB neurons. However, the superplots in the figure seem to show a stark difference in the outcome of the two replicates, with there being no clear suppression of Imp decondensation in one of them. This experiment needs repeating two or three more times to establish which of the outcomes in the current two replicates is robust.

We thank the reviewer for pointing that out. We would however like to clarify that even if the suppression was milder for the first replicate (in red), it was still significant (*P*<0.01 in a Mann-Whitney test comparing control and Kir2.1 conditions in response to Tyramine). As this experiment was the only one performed “only” twice, we launched two new experimental replicates. As shown in the revised Figure 1figure supplement 1D, our four replicates indicate that blocking the firing of MB neurons significantly inhibits Tyramine-dependent RNP granule remodeling. Variations in the efficiency of inhibition are observed between replicates, which might result from small changes in temperature (as the Kir2.1 construct is expressed *via* the Gal4/UAS system) and/or differential responses of MB neurons.

2. In the absence of new functional data, it would be helpful to include some more speculation about how the processes uncovered in this work might relate to tyramine's known functions in the *Drosophila* brain. My understanding is that tyramine has to date been mostly implicated in courtship behaviour. Is there any evidence that courtship paradigms studied in previous work on tyramine involve the studied MB neurons or mRNAs?

We thank the reviewer for the suggestion. We have implemented the Discussion section (page 12) so as to i- better refer to the few physiological functions attributed to Tyramine so far, ii- highlight functions that might be related to MBs, and iii- specify that Tyramine’s functions have so far not been linked to posttranscriptional regulatory mechanisms.

Minor comments:1. The conclusion that decondensation of granules is reversible would be further supported by showing that this is the case for not just Imp (Figure S1D) but also Me31B.

We agree, this experiment has been performed (see our answer to the first point of reviewer 1) and the results are displayed in our revised Figure 1—figure supplement 1C.

2. I was curious why puromycin was used for some experiments and anisomycin for others. These molecules have very different effects on the translational machinery so some further explanation might be helpful.

The reviewer is right in pointing out that puromycin and anisomycin have different effects on the translation machinery. While anisomycin prevents tRNA recruitment by binding to the ribosomal A-site cleft, puromycin enters the ribosomal A site as a tRNA analog and gets covalently coupled to the nascent peptide chain at the P site, thus terminating protein synthesis. Despite this different mode of action, both anisomycin and puromycin block peptide chain elongation and were used interchangeably as translation inhibitors in various systems. In this study, we initially used anisomycin to block translation of reporters RNAs as it had previously been shown to block translation in the context of *Drosophila* brain explants (Ashraf et al., 2006) and to work efficiently at lower concentrations than puromycin (Sidhu and Omiecinski, 1998). While we could have used anisomycin for our subsequent SunTag experiments, we switched to puromycin as this molecule has been used as a gold standard in all articles reporting SunTag experiments to demonstrate the specificity of SunTag translation foci (Pichon et al., 2016; Yan et al., 2016; Wu et al., 2016; Boersma et al., 2019; Cioni et al., 2019; Dufourt et al., bioRxiv 2020).

3. The label to Figure 4A indicates that RNase was present only in the input for the GFP only sample. Presumably this is a labelling error?

We thank the reviewer for pointing this labeling error out. In this assay, the initial lysate (input) was split into two halves, one of them only was treated with RNAse prior to immuno-precipitation. We have removed indication of RNAse treatment in the input fraction and have clarified the procedure used in the Materials and Method section.

4. There are a couple of typos/grammatical errors: (i) bottom of page 9, "with a kinetics" should be "with kinetics; (ii) page 11, "CamkII may thus either phosphorylates Imp"…use singular "phosphorylate".

We thank the reviewer for pointing out these errors that we corrected.

Reviewer #3:Here the authors studied the process of RNP granule remodeling in *Drosophila* neurons. They used high resolution imaging of brain explants to decipher how RNP assemblies respond to different neurotransmitters and neuromodulators. One of these neuromodulators, Tyramine, acts via the TyrR receptor to induce the reversible dissociation of RNP granules that contain the RNA-binding protein Imp and the DEAD box helicase Me31B. Tyramine stimulation has a differential effect on these components – a significant release of Imp, but only a partial relocalization of granular Me31B to the cytoplasm – suggesting that in these RNP assemblies, Me31B acts as a scaffold and Imp as a client. Moreover, they showed that to activate the RNP granule remodelling, a calcium-activated kinase CamkII physically interacts with Imp, which triggers the relocation of Imp and associated mRNAs from granules to the cytoplasm. This results in the translation activation of these mRNAs. Importantly, to demonstrate this, the authors used a new method, the SunTag amplification system, which allowed the high resolution monitoring of the dynamics of the process. In summary, this work contributes to our understanding of how the neuronal stimuli-dependent remodelling of RNP condensates affects the expression of neuronal mRNA.This is a well-written paper with clear and convincing results. The key conclusions are well justified. The experiments are sufficiently replicated and statistical analyses are adequate.Significance:This research would be beneficial for neurobiologists and the scientists studying RNP granules to better understand the effects of neuromodulators and neurotransmitters on these granules and to understand the dynamics and consequences of RNP granule disassembly. In addition, this work lays the foundation for monitoring translation with high spatio-temporal resolution using the Suntag methodology.Comments:Overall, the figures and the text are clear and well-written, although there are a few suggestions:A final, careful proofreading is suggested; we found a very small number of errors (e.g., "pharmacology" should be "pharmacologically"; "CamkII may thus either phosphorylates Imp through non-canonical sites…" – should be "phosphorylate").

We thank the reviewer for pointing out these typos and have corrected them.

Prior knowledge regarding Tyramine and its function can be more elaborated and emphasized.

We thank the reviewer for the suggestion. We have now implemented our discussion so as to better highlight the few described functions of Tyramine (page 12).

Ala peptide and its mechanism to inhibit CamkII can be explained more.

We have now added a sentence describing the properties of the ala peptide (page 8).

Figure 1E might require a different statistical test, as the current test indicates that the data are highly statistically significant, but the decrease in me31B partition coefficient is minimal.

We agree with the reviewer that the high significance reported in Figure 1E (now Figure 1F) is mainly due to the high number of data points and that this may be misleading. The t-test performed in this Figure is however adapted to both sample size and distribution. Still, to take into consideration the reviewer’s comment, we now indicate in the Figure legend that the *P* value calculated when comparing the distributions of replicate means is higher than 0.5 (which then is due to the fact that the number of replicates is 3).

On page 9, the last line of the first paragraph refers to Figure 4C which should actually be Figure 4D.

The reviewer is correct, we have changed 4C into 5C (as 4D moved to 5C).

The meaning and significance of Figure 5E is not very clear and can be explained more.

These plots were aimed at highlighting the match between the dynamics of Tyramine-induced calcium transients and that of SunTag foci burst. In the revised version of the manuscript, we added a sentence to better explain this point (page 9). We also clarified the corresponding legend.

The authors image RNP granules in what they state are cell bodies of MB γ neurons, but it is not clear how they distinguish this population from all the other neurons in the vicinity. Did they use reporter expression (not shown in the manuscript)? Are these neurons recognizable from brain architecture/location in the brain alone? The authors should make clear how they know that these are γ neurons.

Although it is not apparent in our Figure panels, DAPI was used in all our experiments on fixed samples to locate the cell bodies of MB neurons. In these experiments, DAPI staining was combined with labeling of Imp, which is differentially expressed in MB sub-populations, and is in particular absent from αβ neurons, thus enabling recognition of MB γ neurons. Suntag live-imaging experiments were performed using VT44966-Gal4, which is a γ neuron-specific driver. The only experiments in which MB γ neurons were not unambiguously identified are the live-imaging experiments displayed in Figure 2 (Me31-GFP live-imaging). In these experiments, however, Me31B-GFP granules behaved similarly in all MB neurons. This information is now mentioned in our revised Materials and methods section and in the legend to Figure 2.

There are few suggestions regarding the presentation of the data and conclusions:To eliminate any chances of confusion, which color channel corresponds to which protein staining should be clearly stated in all the overlay figures (1A', 1B', S3C').

The color channel attributed to each protein is now indicated in the overlay images of Figures 1A’,B’ and S3C’.

Figure S1A could be included in the main figures.

This has been done (see revised Figure 1).

A schematic representing the overall mechanism described in the paper should be added.

We have now included in Figure 6—figure supplement 1 a model summarizing the results described in our manuscript.

[Editors' note: further revisions were suggested prior to acceptance, as described below.]

The manuscript has been improved but there are some remaining issues that need to be addressed, as outlined below:1. The experiments presented do not seem to show that the axonal / synaptic mRNP granules are altered by tyramine. Instead they show tyramine regulation of somatic granules, which may be reasonable, easily assayable surrogates for synaptic granules. While this is acceptable, it is recommended that this experimental gap be clearly acknowledged, because granules in neuronal processes are most likely the ones important to regulate for memory-related processes.

Whether synaptic mRNP granules are altered by Tyramine is indeed a question that was not addressed in this manuscript. To make it clearer, we have revised the text so as to emphasize that the granules we have been analyzing are found in MB cell bodies (see edits in the abstract, pages 4, 11). We have also modified the last part of our discussion (page 13) to highlight that looking at local regulation represents a future line of research.

2. Please consider if the language used to describe the association of Imp and CamKII is optimal. Particularly, whether terms such as 'partner' and 'interaction' to discuss this issue may be subject to misinterpretation. It could be better to state that these proteins are found in the same complex etc.

Although we agree that the term “partner”, frequently used in the context of complex purification, might be misleading, we don’t see why the term “interaction” or “interactor” should be avoided. As stated in all method textbooks (*e.g.* Protein-Protein Interactions book, Methods Mol Biol series), indeed, coimmunoprecipitation is a technique used to probe for protein-protein interactions (which can be direct or not). In the revised manuscript, we have thus removed the term partner, and have used either “interaction” or “association”.

3. Consider including citations to what appear to be relevant previous work. In particular Barbee et al. 2006 (10.1016/j.neuron.2006.10.028) which is probably the first report of Imp on a class of neuronal granule; and Bakthavachalu et al. 2018 (10.1016/j.neuron.2018.04.032), which describes experiments that perhaps most definitively argue for a function for neuronal RNP granules (and by inference their regulation) in long-term memory.

We thank the editor for this suggestion and have inserted the corresponding references (page 4 and page 13).